

# A dimensionless approach for the runoff peak assessment: effects of the rainfall event structure

Ilaria Gnecco[1], Anna Palla[1], Paolo La Barbera[1]

[1]Department of Civil, Chemical and Environmental Engineering, University of Genova, Genoa, 16145, Italy

*Correspondence to*: Anna Palla (anna.palla@unige.it)

**Abstract.** The present paper proposes a dimensionless analytical framework to predict the hydrologic response of a given catchment thus assessing the impact of the rainfall event structure on the runoff peak. The dimensionless form of the rainfall depth is described as a simple power function of the dimensionless duration. Soil abstractions are modelled using the Soil Conservation Service method and the Instantaneous Unit Hydrograph theory is undertaken to determine the dimensionless

form of the hydrograph; the two-parameter gamma-distribution is selected to test the proposed methodology. A set of analytical expressions are derived in case of constant-intensity hyetograph to assess the highest runoff peak with respect to a given rainfall event structure irrespective of the specific catchment. Looking at the results, the curve of the highest values of the runoff peak reveals a local minimum point in the neighbourhood of $d_*$ and $n$ values equal to 1 and 0.3, respectively. As an example, the proposed approach has been applied to analyse the hydrologic response of a small Mediterranean catchment

to three observed rainfall events characterized by different rainfall internal structures.

## 1 Introduction

The ability to predict the hydrologic response of a river basin is a central feature in hydrology. For a given rainfall event, estimating rainfall excess and transforming it to runoff hydrograph is an important task for planning, design and operation of water resources systems. For these purposes, design storm based on the statistical analysis of the annual maximum series of

rainfall depth are used in practice as input data to evaluate the corresponding hydrograph for a given catchment. Several models are documented in the literature to describe the hydrologic response (e.g. Chow et al., 1988, Beven, 2012): the simplest and most successful is the unit hydrograph concept proposed firstly by Sherman (1932). Due to a limited availability of observed streamflow data mainly in small catchment, the attempts in improving the peak flow predictions are documented in the literature since the last century (e.g. Henderson, 1963; Meynink and Cordery, 1976) to date. Recently,

Rigon et al. (2011) investigated the dependence of peak flows on the geomorphic properties of river basins. In the framework of flood frequency analysis, Robinson and Sivapalan (1997) presented an analytical description of the peak discharge irrespective of the functional form assumed to describe the hydrologic response. Goel et al. (2000) combine a stochastic rainfall model with a deterministic rainfall-runoff model to obtain a physically based probability distribution of flood discharges; results demonstrate that the positive correlation between rainfall intensity and duration impacts the flood





flow quantiles. Vogel et al. (2011) developed a simple statistical model in order to simulate observed flood trends as well as the frequency of floods in a nonstationary word including changes in land use, climate and water uses. Iacobellis and Fiorentino (2000) proposed a derived distribution of flood frequency identifying the combined role played by climatic and physical factors at the catchment scale. Bocchiola and Rosso (2009) developed a derived distribution approach for flood

prediction in poorly gauged catchments to shift the statistical variability of rainfall process into its counterpart in terms of statistical flood distribution.

In this framework, the present research study takes a different approach by exploring peak flow rate values, which are subject to a very broad range of climatic, physical, geomorphic and anthropogenic factors, limited to the rainfall input neglecting the expected rainfall event features condensed in the Depth-Duration-Frequency (DDF) curves. The main focus of

this paper is to assess the impact of the rainfall event structure on the peak flow rate based on a deterministic event-based analysis. With this aim, rainfall-runoff processes are modelled using the Soil Conservation Service (SCS) method for soil abstractions and the Instantaneous Unit Hydrograph (IUH) theory to provide a dimensionless analytical expression for peak flow.

The first specific objective is to define a structure relationship of the rainfall event in terms of a simple power function. The

second specific objective is to analytically derive the highest peak flow rate caused by a rainfall event with given internal structure irrespectively of the specific features of the catchment. Finally, as an example, the proposed approach has been applied to analyse the hydrologic response of a small Mediterranean catchment to three observed rainfall events characterized by different rainfall internal structures.

## 2 Methodology

A dimensionless approach is proposed in order to define an analytical framework that can be applied to any study case (i.e. natural catchment). It follows that both the rainfall depth and the rainfall-runoff relationship that are strongly related to the climatic and morphologic characteristics of the catchment, are expressed through dimensionless forms.

### 2.1 The dimensionless form of the rainfall structure relation

Rainfall DDF curves are commonly used to describe the maximum rainfall depth as a function of duration for given return

periods. In particular for short durations, rainfall intensity has often been considered rather than rainfall depth, leading to intensity-duration-frequency (IDF) curves (Borga et al., 2005). Power laws are commonly used to describe DDF curves in Italy (e.g. Burlando and Rosso, 1996) and elsewhere (e.g. Koutsoyiannis et al., 1998).

In the proposed approach, each rainfall event is described in terms of a simple power function similarly to the DDF curves, therefore assuming that the internal structure relationship of a rainfall event can be described as follow:

$h(d) = a' d^n$                                  (1)




where h [L] is the maximum rainfall depth, $a'$ [LT-n] and n [-] are respectively the coefficient and the structure exponent of the power function for a given duration, d [T].

As an example, Fig. 1 describes the internal structure of a rainfall event according to the above illustrated power function. In Fig. 1, the observed rainfall depth (at the top), the observed and evaluated maximum rainfall depths (at the centre), and the corresponding rainfall structure exponent (at the bottom) are reported.

For a given catchment, by assuming a specific return period Tr [T], the reference value of the maximum rainfall depth, hr [L], is defined according to the corresponding DDF curves, as follows:

$$h_r(T_r, t_r) = a(T_r)t_r^{\,b} \tag{2}$$

where $a(T)$ [LT-b] and b [-] are respectively the coefficient and the scaling exponent of the DDF curve while tr [T] is the reference time of the hydrologic response.

Referring to a rainfall duration corresponding to tr, the rainfall depth is assumed equal to the reference value of the maximum rainfall depth. Based on this assumption a relationship between the parameters of the DDF curve and the rainfall structure function can be derived as follows:

$$h(t_r) = h_r(T_r, t_r) \; \rightarrow a't_r^{\,n} = \; a(T_r)t_r^{\,b} \; \rightarrow \frac{a'}{a(T_r)} = \frac{t_r^{\,b}}{t_r^{\,n}} \tag{3}$$

The dimensionless form of the rainfall depth, $h_*$, is defined by the ratio of the rainfall depth to the reference value of the maximum rainfall depth; similarly the dimensionless duration, $d_*$, is expressed by the ratio of the duration to the reference time. Therefore, the dimensionless form of the rainfall structure relationship may be expressed utilizing Eqs. (1), (2) and (3):

$$h_*(d_*) = \frac{h}{h_r} = \frac{a'd^n}{a(T_r)t_r^{\,b}} = \frac{d^n}{t_r^{\,n}} = d_*^{\,n} \tag{4}$$

**2.2 The dimensionless form of the Unit Hydrograph**

The hydrologic response of a river basin is here predicted through a deterministic lumped model: the interaction between rainfall and runoff is analysed by viewing the catchment as a lumped linear system (Bras, 1990). The response of a linear system is uniquely characterized by its impulse response function, called the Instantaneous Unit Hydrograph (IUH). For the IUH, the excess rainfall of unit amount is applied to the drainage area in zero time (Chow et al., 1988). To determine the dimensionless form of the unit hydrograph a functional form for the IUH and thus the S-hydrograph has to be assumed. In this paper the IUH shape is described with the two-parameter gamma distribution (Nash, 1957):

$$f(t) = \frac{1}{k\Gamma(\alpha)}\left(\frac{t}{k}\right)^{\alpha-1} e^{-\left(\frac{t}{k}\right)} \tag{5}$$

where $f(t)$ [T$^{-1}$] is the IUH, $\Gamma$ [-] is the gamma function, α [-] is the shape parameter while $k$ [T] is the scale parameter. In the well-known two-parameter Nash model, the parameters $\alpha$ and $k$ represent the number of linear reservoirs added in series





and the time constant of each reservoir, respectively. The product $\alpha k$ is the first order moment thus corresponding to the mean lag time of the IUH. Note that the IUH parameters can be related to watershed geomorphology; in these terms the Geomorphologic Unit Hydrograph (GUH) theory attempts to relate the IUH of a catchment to the geometry of the stream network (e.g. Rodriguez-Iturbe and Valdes, 1979; Rosso, 1984).

The dimensionless form of the IUH is obtained by using the dimensionless time, $t_*$, defined as follows:

$$t_* = \frac{t}{\alpha k} \tag{6}$$

The proposed dimensionless approach is based on the use of the IUH scale parameter as the reference time of the hydrologic response (i.e. $t_r = \alpha k$). Using the first order moment in the dimensionless procedure, the approach can be applied to any IUH form. By applying the change of variable $t = \alpha k\, t_*$, the IUH may be expressed as follows:

$$f(t) = \frac{1}{k \Gamma(\alpha)} \left( \frac{\alpha k\, t_*}{k} \right)^{\alpha-1} e^{-\left( \frac{\alpha k\, t_*}{k} \right)} \tag{7}$$

The dimensionless form of the IUH, $f(t_*)$, is defined and derived from Eq. (7) as follows:

$$f(t_*) = f(t) \cdot \alpha k = \frac{\alpha}{\Gamma(\alpha)} (\alpha t_*)^{\alpha-1} e^{-(\alpha t_*)} \tag{8}$$

Note that for the dimensionless IUH the first order moment is equal to one and the time-to-peak can be expressed as follows:

$$\frac{df(t_*)}{dt_*} = 0 \;\; \rightarrow \;\; t_{p*} = \frac{\alpha-1}{\alpha} \tag{9}$$

The dimensionless Unit Hydrograph (UH) is derived by integrating the dimensionless IUH:

$$S(t_*) = \int_0^{t_*} f(\tau_*) d\tau_* \tag{10}$$

where $S(t_*)$ is the dimensionless S-curve (e.g. Henderson, 1963).

For a unit dimensionless rainfall of a given dimensionless duration, $d_*$, the dimensionless UH is obtained by subtracting the two consecutive S curves that are lagged $d_*$:

$$U(t_*) = \begin{cases} S(t_*) & for\ t_* < d_* \\ S(t_*) - S(t_* - d_*) & for\ t_* \geq d_* \end{cases} \tag{11}$$

where $U(t_*)$ is the dimensionless UH.  The time-to-peak of the dimensionless UH, $t_{p*}$,  is derived by solving $dU(t_*)/dt_* = 0$ . Using (8) and (11) and recognizing that $t_{p*} \geq d_*$ gives the following equation for $t_{p*}$:

$$f(t_{p*}) = f(t_{p*} - d_*) \;\rightarrow\; t_{p*} = d_* \frac{e^{\frac{\alpha d_*}{\alpha-1}}}{e^{\frac{\alpha d_*}{\alpha-1}} - 1} \tag{12}$$

Similar expressions for the time-to-peak are available in the literature (e.g. Rigon et al., 2011; Robinson and Sivapalan, 25    1997). Consequently the peak value of the dimensionless UH may be expressed as a function of $d_*$ by:





$$U_{max}(d_*) = S(t_{p*}) - S(t_{p*} - d_*) \tag{13}$$

## 2.3 The dimensionless runoff peak analysis

Based on the unit hydrograph theory and assuming a rectangular hyetograph of duration $d_*$, the dimensionless convolution equation for a given catchment becomes:

$$Q(t_*) = i_e(d_*)U(t_*) \tag{14}$$

where $Q(t_*)$ is the dimensionless hydrograph and $i_e(d_*)$ is the dimensionless excess rainfall intensity.

In the following sections the dimensionless hydrograph and the corresponding peak are examined in case of constant and variable runoff coefficients.

### 2.3.1 The analysis in case of constant runoff coefficient

By considering a constant runoff coefficient, $\varphi_0$, similarly to the dimensionless rainfall depth $h_*$ the dimensionless excess rainfall depth $h_{e*}$ is defined by:

$$h_{e*} = \frac{\varphi_0 h}{\varphi_0 h_r} = d_*^{\,n} \tag{15}$$

The corresponding dimensionless excess rainfall intensity becomes:

$$i_{e*} = d_*^{\,n-1} \tag{16}$$

From Eqs. (13), (14) and (16), the dimensionless hydrograph and the corresponding peak may be expressed by:

$$Q(t_*) = d_*^{\,n-1} U(t_*) \tag{17}$$

$$Q_{max}(d_*) = d_*^{\,n-1} U_{max}(d_*) = d_*^{\,n-1}[S(t_{p*}) - S(t_{p*} - d_*)] \tag{18}$$

In order to investigate the critical condition for a given catchment which maximizes the runoff peak, the partial derivative of the Eq. (18) with respect to the variable $d_*$ is calculated.

$$\frac{\partial Q_{max}(d_*)}{\partial d_*} = 0 \quad \rightarrow \quad \frac{f(t_{p*})d_*}{1-n} = S(t_{p*}) - S(t_{p*} - d_*) = U_{max}(d_*) \tag{19}$$

The analytical expression for estimating the critical duration of rainfall that maximizes the peak flow was firstly derived by Meynink and Cordery (1976).

From Eq. (19) it is possible to derive the $n$ structure value that maximizes the dimensionless runoff peak for a specific duration $d_*$ referring to a given catchment:

$$n = 1 - \frac{f(t_{p*})d_*}{U_{max}(d_*)} \tag{20}$$




### 2.3.2 The analysis in case of variable runoff coefficient

In order to take into account the variability of the infiltration process during the rainfall event, a variable runoff coefficient, $\varphi$, is introduced. The variable runoff coefficient is estimated based on the SCS method for computing soil abstractions (SCS, 1985). Since the analysis deals with high rainfall intensity events it would be reasonable to force the SCS-method in order to

always produce runoff (Boni et al., 2007). The assumption that the rainfall depth always exceeds the initial abstraction is implemented in the model by supposing that a previous rainfall depth at least equal to the initial abstraction occurred; therefore, the excess rainfall depth $h_e$ is evaluated as follows:

$$h_e = \varphi h = \frac{h^2}{h+S} \rightarrow \varphi = \frac{h}{h+S} \tag{21}$$

where $S$ is the soil abstraction [L]. The variable runoff coefficient is therefore described as a monotonic increasing function

of the rainfall depth. It follows that the runoff component is affected by the variability of the infiltration process: the runoff is reduced in case of small rainfall events and is enhanced in case of heavy events.

The dimensionless excess rainfall depth, $h_{e*}$, is defined by:

$$h_{e*} = \frac{h_e}{h_{e_r}} = \frac{\varphi h}{\varphi_r h_r} = \frac{\varphi}{\varphi_r} h_* = \frac{\varphi}{\varphi_r} d_*{}^n \tag{22}$$

The corresponding dimensionless excess rainfall intensity becomes:

$$i_{e*} = \frac{\varphi}{\varphi_r} d_*{}^{n-1} \tag{23}$$

From Eq. (21) the ratio $\frac{\varphi}{\varphi_r}$ may be determined in terms of $h_*$:

$$\frac{\varphi}{\varphi_r} = \frac{h/(h+S)}{h_r/(h_r+S)} = h_* \left(\frac{h_r+S}{h+S}\right) = h_* \left(\frac{1+S_*}{h_*+S_*}\right) \tag{24}$$

where $S_*$ is the dimensionless soil abstraction defined by the ratio of $S$ to $h_r$. The ratio $\frac{\varphi}{\varphi_r}$ is lower than one when the dimensionless rainfall depth is lower than one and vice versa. In the domain of $h_* < 1$ (i.e. $d_* < 1$), the variable runoff

coefficient implies that the runoff component is reduced with respect to the reference case and vice versa. The impact of the ratio $\frac{\varphi}{\varphi_r}$ on the runoff production is enhanced if $S_*$ increases thus causing a wider range of runoff coefficients.

From Eqs. (13), (14) and (23), the dimensionless hydrograph and the corresponding peak may be expressed by:

$$Q(t_*) = \frac{\varphi}{\varphi_r} d_*{}^{n-1} U(t_*) \tag{25}$$

$$Q_{max}(d_*) = \frac{\varphi}{\varphi_r} d_*{}^{n-1} U_{max}(d_*) = \frac{\varphi}{\varphi_r} d_*{}^{n-1} [S(t_{p*}) - S(t_{p*} - d_*)] \tag{26}$$





Similarly to the runoff peak analysis carried out in case of the constant runoff coefficient, the partial derivative of the Eq. (26) with respect to the variable $d_*$ is calculated:

$$\frac{\partial Q_{max}(d_*)}{\partial d_*} = 0 \; \rightarrow \; f(t_{p*})d_* = \left[S(t_{p*}) - S(t_{p*} - d_*)\right]\left[1 - 2n + \frac{nd_*^n}{d_*^n + S_*}\right] \tag{27}$$

From Eq. (27) it is possible to implicitly derive the $n$ structure value that maximizes the dimensionless runoff peak for a specific duration $d_*$ referring to a given catchment.

## 3 Results and discussion

The proposed dimensionless approach is tested using the two-parameter gamma distribution for the shape parameter equal to 3. Such assumption is derived by using the Nash model relation proposed by Rosso (1984) to estimate the shape parameter based on Horton order ratios according to which the $\alpha$ parameter is generally in the neighbourhood of 3 (La Barbera and Rosso, 1989; Rosso et al., 1991). In Fig. 2, the dimensionless rainfall duration is plotted vs. the dimensionless time-to-peak together with the dimensionless IUH and the corresponding dimensionless UH for $d_*=1.0$. Note that the dotted grey lines indicates the UH peak while the dashed grey lines show $t_{p*}$, $f(t_{p*})$ and $f(t_{p*} - d_*)$, respectively.

The dimensionless UH is evaluated varying the dimensionless rainfall duration; then the runoff peak analysis is carried out in case of constant and variable runoff coefficients. Finally a numerical example of the application to a small Mediterranean catchment is presented.

In the following sections the achieved results are presented with respect to the dimensionless durations in the range between 0.5 and 2 that is wide enough to include the duration of the rainfall able to generate the maximum peak flow for a given catchment (Robinson and Sivapalan, 1997).

### 3.1 Highest dimensionless runoff peak with constant runoff coefficient

The dimensionless form of the hydrograph is shown in Fig. 3 with varying the rainfall structure exponents, $n$, for the selected dimensionless rainfall duration. The hydrographs are obtained for excess rainfall intensities characterized by constant runoff coefficient and rainfall structure exponents of 0.2, 0.3, 0.5 and 0.8.

The impact of the rainfall structure exponents on the hydrograph form depends on the rainfall duration: for $d_*$ lower than one, the higher $n$ the lower is the peak flow rate and vice versa.

Figure 4 illustrates the contour plot of the dimensionless runoff peak as a function of the rainfall structure exponent and the dimensionless rainfall duration. In the contour plot, it is possible to observe a saddle point located in the neighbourhood of $d_*$ and $n$ values equal to 1 and 0.3, respectively. Note that the intersection line (reported as bold line in Fig. 4) between the saddle surface and the plane of the principal curvatures where the saddle point is a minimum indicates the highest values of the runoff peak for a given $n$ structure exponent.



In Fig. 5, the highest dimensionless runoff peak and the corresponding rainfall structure exponent are plotted vs. the dimensionless time-to-peak. Further, the dimensionless IUH and the corresponding dimensionless UH for $d_*$=1.0 are reported as an example. The reference line (short-short-short dashed grey line) indicates the lower control line corresponding to the rainfall duration infinitesimally small. Note that the rainfall structure exponent that maximizes the runoff peak for a given duration can be simply derived as a function of the dimensionless time-to-peak (see Eq. 20). The highest dimensionless runoff peak tends to one for long dimensionless rainfall duration ($d_* > 4$) when consequently the $n$ structure exponent tends to one (see Eq. 18). Results confirm that the highest runoff peak curve reveals the local minimum point at $t_{p*}$ of 1.29 corresponding to $n$ of 0.31 and $d_*$ of 1. In light of such trend, it emerges that the less critical runoff peak occurs at $n$ structure exponent values corresponding to the ones typically derived by the statistical analysis of the annual maximum rainfall depth series in Mediterranean climate. In other words, referring to the Chicago hyetograph commonly used in the engineering practice as design storm (Kiefer and Chu, 1957), results illustrated in Fig. 5 reveal that although Chicago hyetograph shows the maximum intensity over each duration, such rainfall condition may not be representative of the most critical condition in terms of runoff peak for a given catchment at assigned return period. At the same time, looking at the highest runoff peak curve there are different rainfall event conditions (rainfall structure exponent $n$ and duration $d$) in the neighborhood of the minimum point that determine comparable effects in term of the runoff peak value. Note that these comparable effects are related to rainfall depths with different return periods for given durations.

**3.2 Highest dimensionless runoff peak with variable runoff coefficient**

The excess rainfall depth, in the case of variable runoff coefficient, is evaluated by assigning a value to the reference runoff coefficient. In particular, the reference runoff coefficient is defined as follows utilizing Eq. (21):

$$\varphi_r = \frac{h_r}{h_r + S} \rightarrow \varphi_r = \frac{1}{1 + S_*} \tag{28}$$

In order to provide an example of the proposed approach, the presented results are obtained assuming a dimensionless soil abstraction $S_*$ of 0.25. It follows that the reference runoff coefficient $\varphi_r$ is equal to 0.8.

Similarly to the results presented for the case of constant runoff coefficient, Fig. 6 illustrates the dimensionless hydrographs obtained for excess rainfall intensities characterized by variable runoff coefficient and $n$ structure exponents of 0.2, 0.3, 0.5 and 0.8. at assigned dimensionless rainfall duration ($d_*$=0.5, 1.0, 1.5, and 2.0). The dimensionless hydrographs, obtained for the variable runoff coefficient, show the same behaviours of the ones derived for the constant runoff coefficient (see Figs. 3 and 6), even if they differ in magnitude, thus confirming the role of the variable runoff coefficient on the runoff peak. In particular, due to the variability of the infiltration process, the runoff peaks slightly decrease for rainfall duration lower than one (i.e. $d_*$=0.5) when compared with the ones observed in case of constant runoff coefficient while they rise up for duration larger than one (i.e. $d_*$=1.5 and 2).



Figure 7 shows the contour plot of the dimensionless runoff peak as a function of the rainfall structure exponent and the dimensionless rainfall duration in case of variable runoff coefficient. By comparing Figs. 7 and 4, it emerges that the contour lines observed in case of variable runoff coefficient reveal a steeper trend with respect to constant runoff coefficient ones indeed the impact of the $n$ structure exponent on the runoff peak is enhanced when the runoff coefficient is assumed variable.

The saddle point is again located in the neighbourhood of $d_*$ and $n$ values equal to 1 and 0.3, respectively while the curve of the highest values of the runoff peak (reported as bold line in Fig. 7) is moved on the left.

In Fig. 8, the highest dimensionless runoff peak and the corresponding rainfall structure exponent are plotted vs. the dimensionless time-to-peak in case of variable runoff coefficient. Results plotted in Fig. 8 confirm that the highest runoff peak curve reveals the local minimum point at $t_{p*}$ of 1.29 corresponding to $n$ of 0.26 and $d_*$ of 1. Referring to $S_*$ of 0.25, the

highest dimensionless runoff peak tends to 1.25 for long dimensionless rainfall duration ($d_* > 4$) when consequently the $n$ structure exponent tends to one (see Eqs. 24 and 26).

Figure 9 illustrates the highest dimensionless runoff peak and the corresponding rainfall structure exponent vs. the dimensionless time-to-peak in case of variable runoff coefficient (for $S_*$ values of 0.25 and 0.67) together with the comparison to the case of constant runoff coefficient. The highest dimensionless runoff peak are similar for short rainfall

duration (i.e. $t_{p*}$ lower than 1.5) when the variable runoff coefficient reduces the runoff component with respect to the reference runoff case (that is also the constant runoff case i.e. $S_*=0$). On the contrary, the highest dimensionless runoff peak increases with increasing the dimensionless soil abstraction for long rainfall duration. Indeed, in these cases, the variable runoff coefficient enhances significantly the runoff component with respect to the constant runoff case (i.e. $S_* = 0$). The rate of change in the runoff production ascribable to the variable runoff coefficient is predominant with respect to the one due to

the rainfall duration increase, therefore the $n$ structure exponent that maximizes the runoff peak, decreases for increasing the dimensionless soil abstractions.

### 3.3 Catchment application

In order to provide a numerical application of the proposed methodology, this approach has been implemented for the Bisagno catchment at La Presa station, located at the centre of Liguria Region, (Genoa, Italy).

The Bisagno – La Presa catchment has a drainage area of 34 km$^2$ with an index flood of about 95 m$^3$/s. The upstream river network is characterized by main channel length of 8.36 km and mean streamflow velocity of 2.4 m/s. Regarding the geomorphology of the catchment, the area ($R_A$), bifurcation ($R_B$) and length ($R_L$) ratios that are evaluated according to the Horton-Strahler ordering scheme, are respectively equal to 5.9, 5.6 and 2.5. By considering the altimetry, vegetation and limited anthropogenic exploitation of the territory, the Bisagno – La Presa is a mountain catchment characterized by an

average slope of 33%. The soil abstraction, $S_{II}$ is assumed equal to 41 mm; its evaluation is based on the land use analysis provided in the framework of the EU Project CORINE (EEA, 2009). The mean value of the annual maximum rainfall depth for unit duration (hourly) and the scaling exponent of the DDF curves are respectively equal to 41.31 mm/h and 0.39.





Detailed hydrologic characterization of the Bisagno catchment can be found elsewhere (Bocchiola and Rosso, 2009; Rulli and Rosso, 2002; Rosso and Rulli, 2002). Focusing on the rainfall-runoff process the two parameters of the gamma distribution are evaluated based on the Horton order ratio relationship (Rosso, 1984). The shape and scale parameters are estimated equal to 3.4 and 0.25 h respectively, thus corresponding to the lag time of 0.85 h.

In this application three rainfall events observed in the catchment area have been selected in order to analyse the different runoff peaks occurred for the three rainfall internal structures. The selected events are characterized by analogous magnitude of the maximum rainfall depth observed for the duration equal the reference time (i.e. $h_r = 80$ mm, $t_r = 0.85$ h).

Figure 10 illustrates the internal structure of the three selected rainfall events. The graphs at the top report the observed rainfall depths while the central graphs show the estimated rainfall structure exponents. At the bottom of Fig. 10, by

considering the three structure exponents corresponding to the Bisagno – La Presa reference time (i.e. $n = 0.55, 0.62, 0.71$), the rainfall structure curves are derived for a rainfall durations ranging between $0.5 \cdot t_r$ and $2 \cdot t_r$; for comparison purpose, the DDF curve is also reported. Based on each rainfall structure curve, four rectangular hyetographs with duration of 0.425, 0.85, 1.275, and 1.7 h are derived to evaluate the impact on the hydrologic response of the Bisagno – La Presa catchment. Note that the analysis is performed in case of variable runoff coefficient whose reference value is equal to 0.66 (i.e. $S_* = 0.5$).

In Fig. 11, the net hyetographs, the corresponding hydrographs and the reference value of the runoff peak flow are plotted for the three investigated rainfall structure exponents. The reference value of the runoff peak flow (dash-dot line) is evaluated by assuming a constant-intensity hyetograph of infinite duration and having excess rainfall intensity equal to the one estimated for the reference time. The role of the rainfall structure exponent emerges in the different decreasing rate of the excess rainfall intensity with the duration, thus resulting in the corresponding increasing rate of the peak flow values.

Figure 12 shows the contour plot of the dimensionless runoff peak in case of variable runoff coefficient ($S_* = 0.5$). The highest runoff peak curve is also reported (bold line) together with the dimensionless hydrograph peaks (grey-filled stars) for the selected rainfall structure exponents ($n = 0.55, 0.62, 0.71$) and durations ($d_* = 0.5, 1.0, 1.5,$ and 2.0). Similarly to Fig. 7, the Bisagno – La Presa catchment application shows a curve of the highest values of the runoff peak characterized by a local minimum (saddle point) in the neighbourhood of $d_*$ and $n$ values equal to 1 and 0.3, respectively.

**4 Conclusions**

The proposed analytical dimensionless approach allows predicting the hydrologic response of a given catchment; particular attention has been posed on the assessment of the runoff peak commonly required for design purposes.

Both the rainfall depth and the rainfall-runoff relationships are expressed through dimensionless forms: the first one is described in terms of a simple power function while the SCS method and the IUH theory are undertaken to model the

rainfall-runoff process. The proposed approach is therefore valid within a framework that assumes that the watershed is a linear causative and time invariant system, where only the rainfall excess produces runoff. In the present paper the two-parameter gamma distribution is adopted as IUH form; however the analysis can be repeated using other IUH forms




obtaining similar results. Indeed, as previously addressed by Robinson and Sivapalan (1997) the actual IUH shape is of secondary importance if the main objective is estimating the peak discharge.

A set of analytical expressions has been derived to provide the estimation of the highest peak with respect to a given n structure exponent. Results reveal the impact of the rainfall event structure on the runoff peak thus pointing out the following

features:

- the curve of the highest values of the runoff peak reveals a local minimum point (saddle point);
- different combinations of n structure exponent and rainfall duration may determine similar conditions in terms of runoff peak.

Referring to the Bisagno – La Presa catchment application, the saddle point of the the runoff peaks is located in the

neighbourhood of n value equal to 0.3 and rainfall duration corresponding to the reference time ($d_* = 1$). Further, it emerge that the highest runoff peak value corresponding to the scaling exponent of the DDF curve is comparable to the less critical one (saddle point).

Findings of the present research suggest reviewing the derived flood distribution approaches that coupled the information on precipitation via DDF curves and on the catchment response based on the iso-frequency hypothesis. Future research with

regard to the structure of the extreme rainfall event is needed; in particular the analysis of several rainfall data series belonging to a homogeneous climatic region is required in order to investigate the frequency distribution of specific rainfall structures.

The developed approach, besides suggesting remarkable issues for further researches and unlike the merely analytical exercise succeeds in highlighting once more the complexity in the assessment of the maximum runoff peak.

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





**Figure 1: Internal structure of a rainfall event according to a power law. The observed rainfall depth (at the top), the observed and evaluated maximum rainfall depths (at the centre), and the corresponding rainfall structure exponent (at the bottom) are reported.**




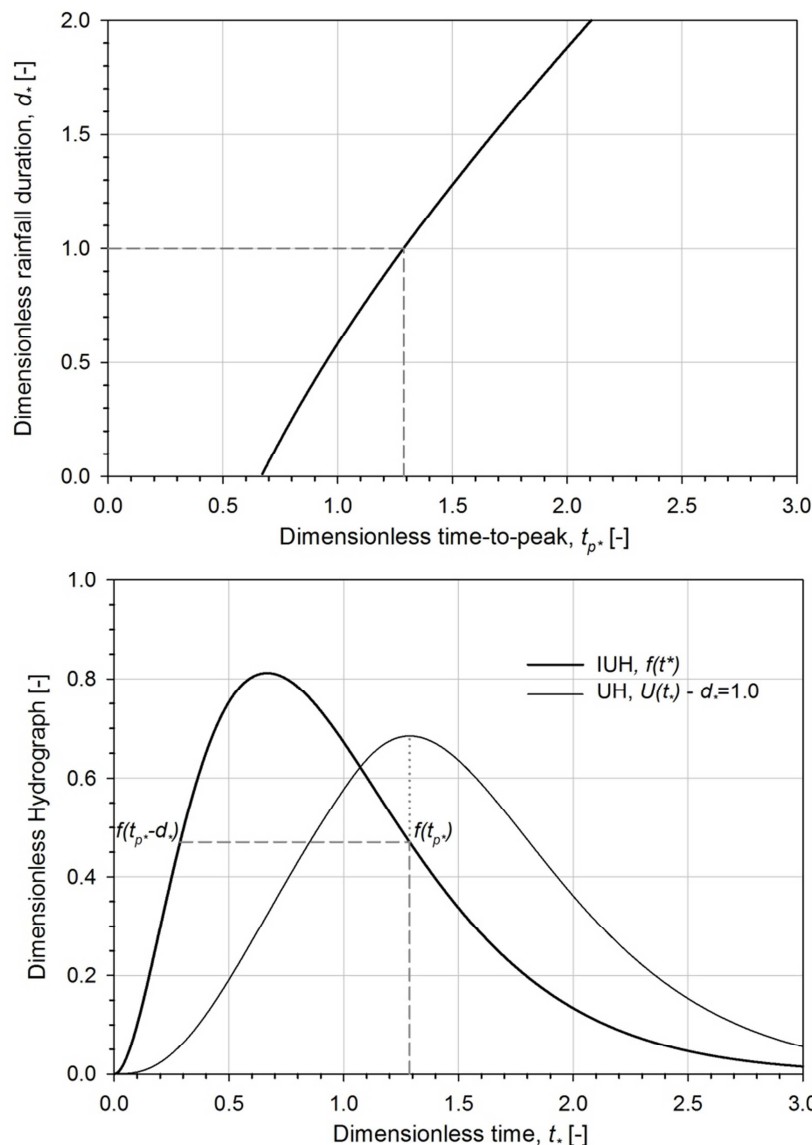

**Figure 2: Dimensionless rainfall duration vs. dimensionless time-to-peak; Dimensionless Instantaneous Unit Hydrograph and the corresponding dimensionless Unit Hydrographs for $d_*$=1.0. Note that the shape parameter α is equal to 3.**





**Figure 3: Dimensionless Hydrographs obtained for excess rainfall intensities characterized by constant runoff coefficient and different rainfall structure exponents, $n$ ($n$ = 0.2, 0.3, 0.4 and 0.5) at assigned dimensionless rainfall duration, $d_*$ ($d_*$=0.5, 1.0, 1.5, and 2.0). Note that the shape parameter $\alpha$ is equal to 3.**




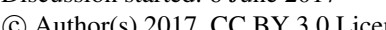

Figure 4: Contour plot of the dimensionless runoff peak as a function of the rainfall structure exponent and the dimensionless rainfall duration in case of constant runoff coefficient. The maximum dimensionless runoff peak curve is also reported (bold line).





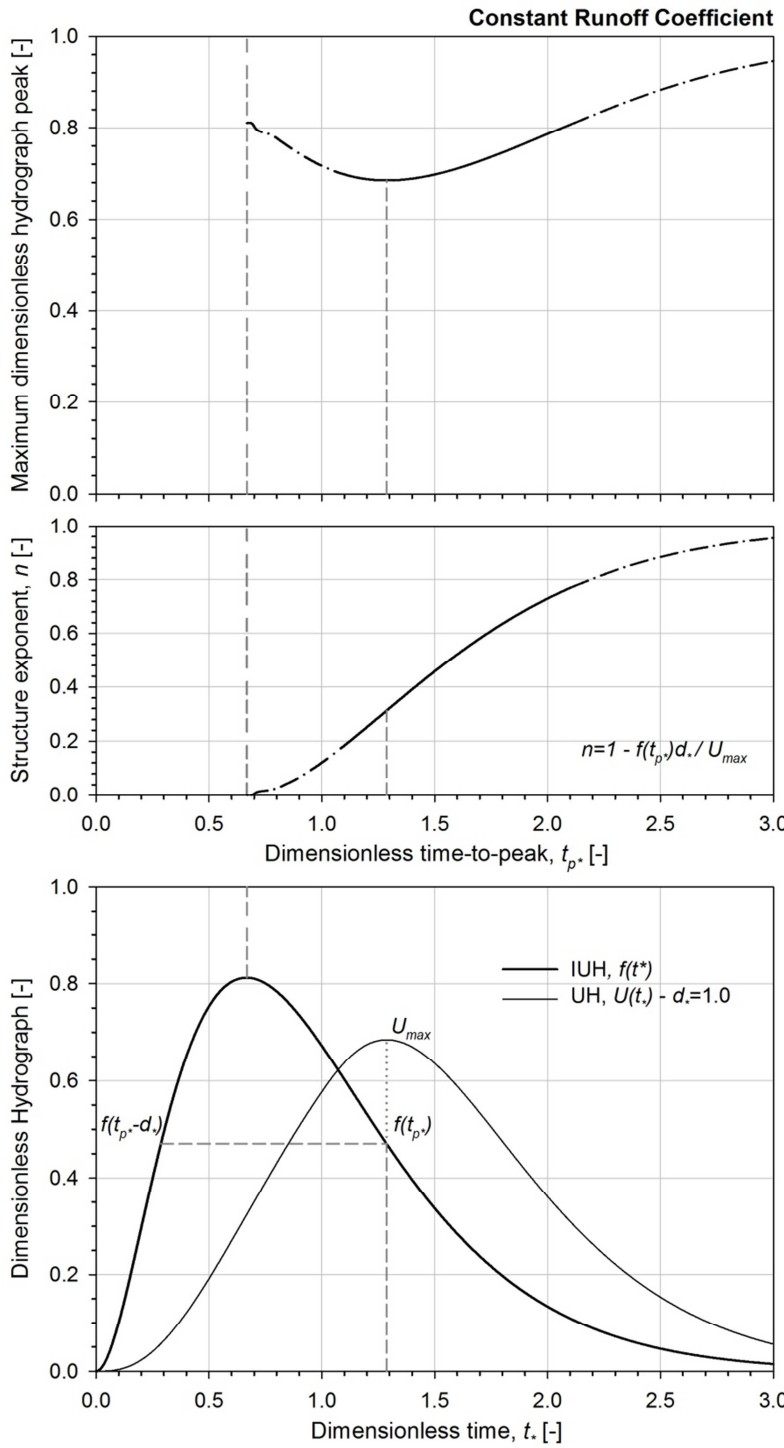

**Figure 5: Maximum dimensionless runoff peak and the corresponding rainfall structure exponent vs. dimensionless time-to-peak in case of constant runoff coefficient; Dimensionless Instantaneous Unit Hydrograph and the corresponding dimensionless Unit Hydrographs for $d_*$=1.0. Note that the shape parameter α is equal to 3.**





**Figure 6: Dimensionless Hydrographs obtained for excess rainfall intensities characterized by variable runoff coefficient and different rainfall structure exponent, _n_ (_n_ = 0.2, 0.3, 0.4 and 0.5) at assigned dimensionless rainfall duration, $d_*$ ($d_*$=0.5, 1.0, 1.5, and 2.0). Note that the shape parameter α is equal to 3.**





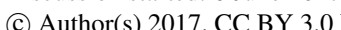

**Figure 7: Contour plot of the dimensionless runoff peak as a function of the rainfall structure exponent and the dimensionless rainfall duration in case of variable runoff coefficient. The maximum dimensionless runoff peak curve is also reported (bold line).**




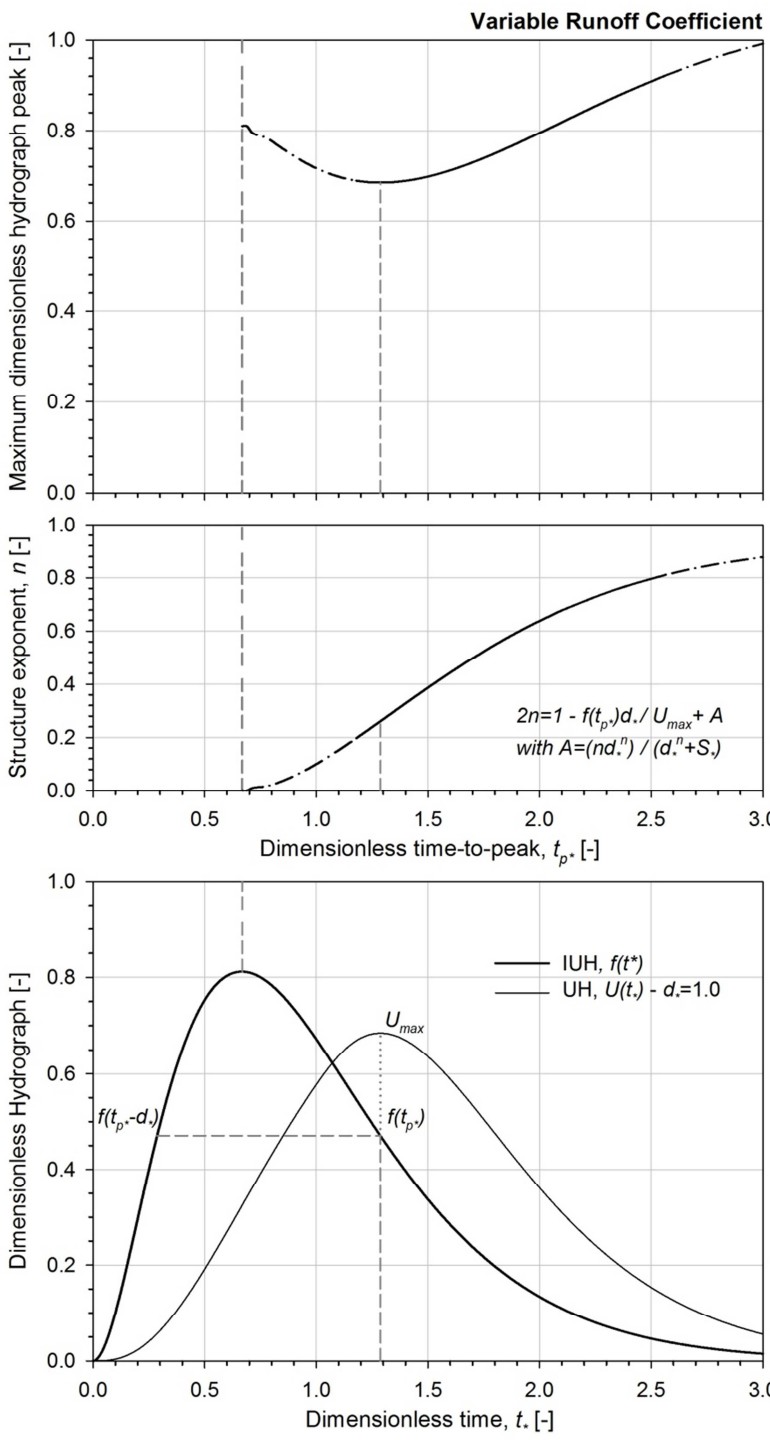

**Figure 8: Maximum dimensionless runoff peak and the corresponding rainfall structure exponent vs. dimensionless time-to-peak in case of variable runoff coefficient; Dimensionless Instantaneous Unit Hydrograph and the corresponding dimensionless Unit Hydrographs for $d_*$=1.0. Note that the shape parameter α is equal to 3.**





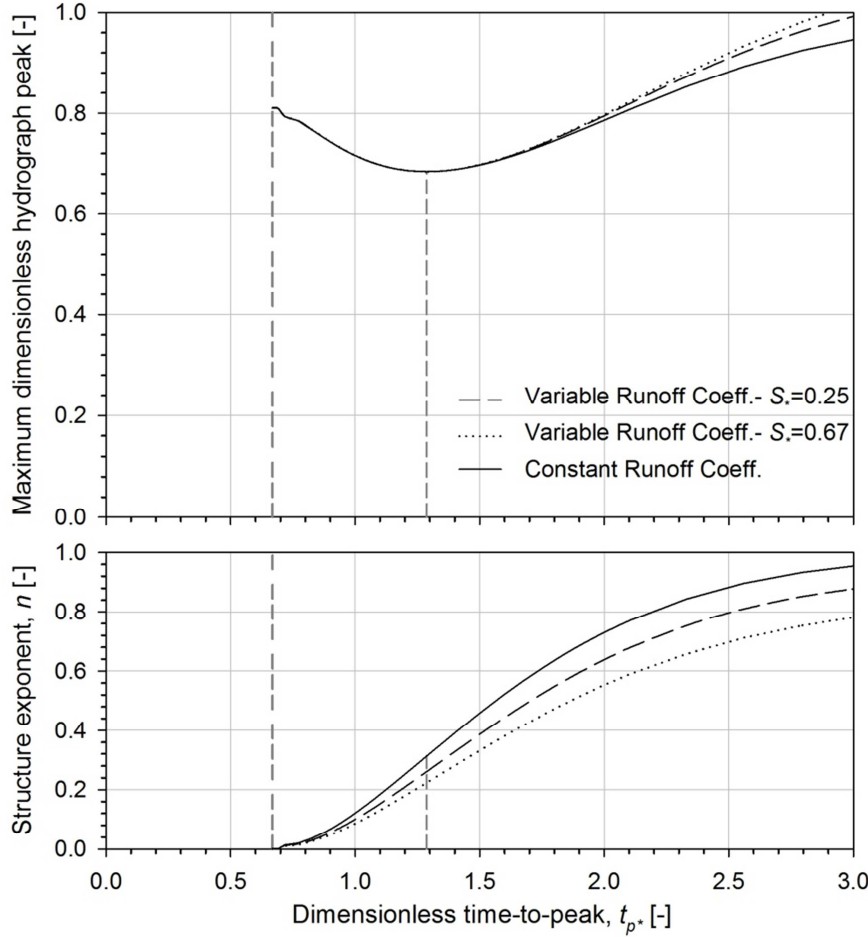

**Figure 9: Maximum dimensionless runoff peak and the corresponding rainfall structure exponent vs. dimensionless time-to-peak in case of variable runoff coefficients with respect to dimensionless maximum retention $S_*$ of 0.25 and 0.67. The comparison to the case of constant runoff coefficient is also reported.**





**Figure 10: Internal structure of three rainfall events observed in Genoa (IT): the observed rainfall depths (at the top) and the estimated rainfall structure exponents (at the centre) are reported. At the bottom, the rainfall structure curves evaluated for the reference time of the Bisagno – La Presa catchment and the corresponding Depth-Duration-Frequency curves are reported.**




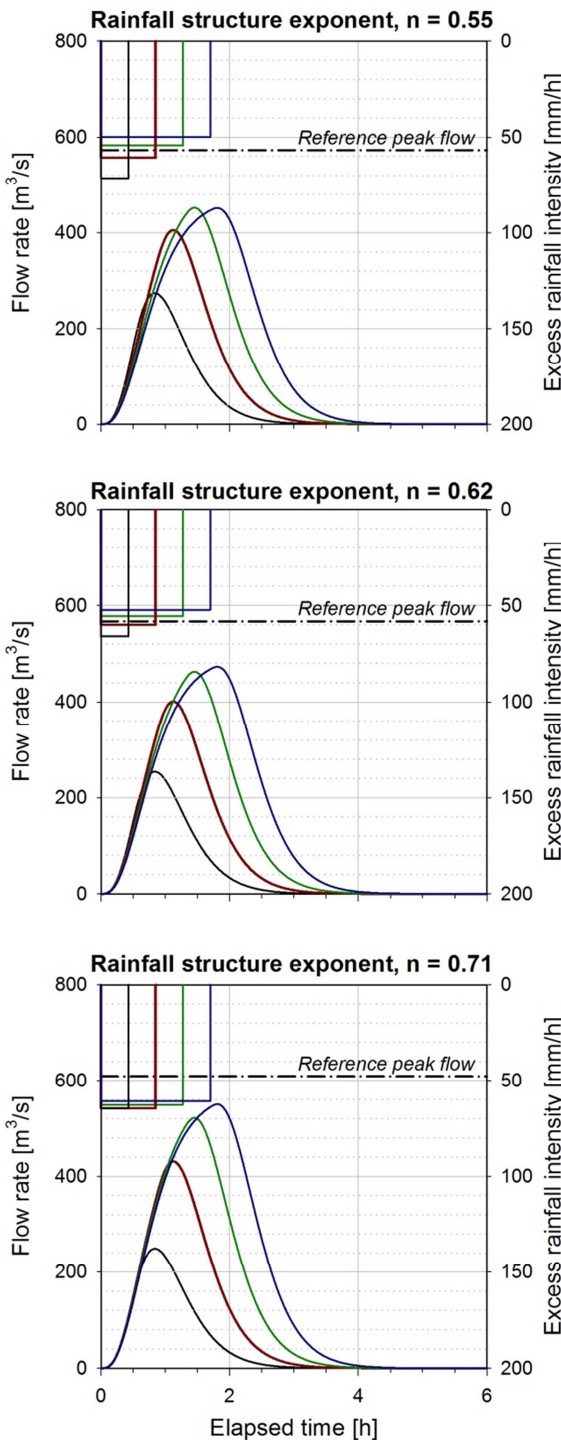

**Figure 11: The excess rainfall hyetographs, the corresponding hydrographs and the reference value of the runoff peak flow for the Bisagno – La Presa catchment evaluated for three rainfall structure exponents. Note that each graph includes four rainfall durations (i.e. 0.5, 1.0, 1.5, and 2.0 times the reference time).**





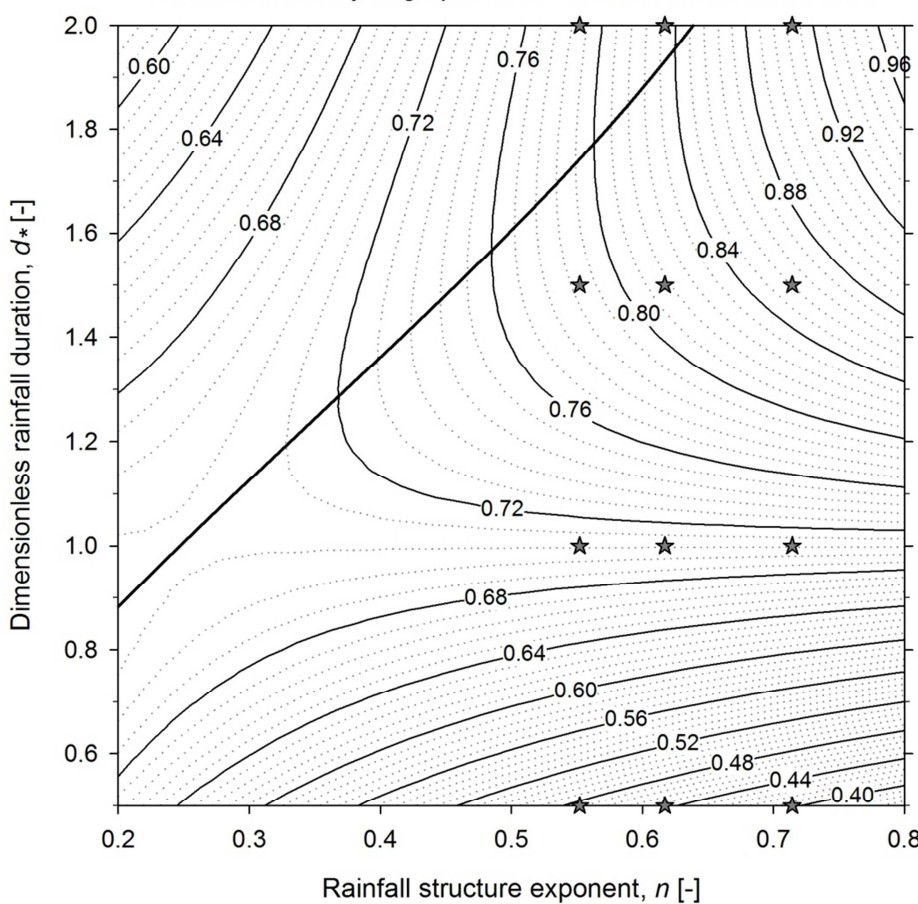

**Figure 12: Contour plot of the dimensionless runoff peak evaluated for the Bisagno – La Presa catchment in case of variable runoff coefficient ( $S_*$ =0.5). The maximum dimensionless runoff peak curve is also reported (bold line) together with the dimensionless hydrograph peaks (grey-filled stars) for the selected rainfall structure exponents ($n$ = 0.55, 0.62, 0.71) and durations ($d_*$= 0.5, 1.0, 1.5, and 2.0).**