# Peer review of "A dimensionless approach for the runoff peak assessment: effects of the rainfall event structure"

_Hydrology and Earth System Sciences, 2017_

## Referee Comment (RC1) · G. Baiamonte (Referee) · 20 Jul 2017

General comments

This paper describes an interesting and heuristic analytical solution for the assessment of the peak runoff. By considering the SCS method and the IUH theory to model the soil abstraction and the hydrograph, respectively, a new approach to assess the peak runoff of a given catchment is presented. Thus, the Authors combine models already known; however, the dimensionless approach they used seems to be new and allows compacting the solution in order to also investigate the interesting feature of the rainfall even structure. Due to personal interest into the concept of peak discharge, I followed

the mathematical derivation and do confirm that it is sound and well described. I can find no flaws in their approach and agree with their findings and conclusions. I found the exposed methodology well developed and sufficiently comprehensive. I am of the opinion that these results are of interest for the readers of HESS, thus the manuscript is suitable for the publication.

Specific comments

The only concern about this paper is the lacking of a comparison of the analytical approach to the hydrological response with some available analytical model previously published in order to deepen its degree of applicability. Otherwise, the main findings could be compared with experimental or numerical data that are available in the literature. This should not be too much time consuming to achieve and it will significantly strengthen the paper. The application for the Bisagno catchment could not be considering as a test of the suggested solution. Therefore, my recommendation is to try to achieve such a task.

It is not clear to me how the rainfall structure exponent in the example of Figure 1 (at the bottom) and in the application for the Bisagno catchment at La Presa station (Figure 10 at the centre) is determined. Although based on previous studies, n exponent determination vs rainfall duration should be described. How does n qualitatively influence the rainfall structure ? I checked Figure 1 at the centre, is the estimated one a a simple power law ? Please, add parameters (the same for Fig. 10).

Figure 4 and 7 are very effective. However, to show the effect of the rainfall structure parameter, it could be useful also plotting the dimensionless hydrograph peak vs rainfall structure exponent, for both constant and time-varying psi. Moreover, A 3D figure could better evidence the influence of the rainfall structure on the dimensionless peak discharge and the saddle area. The actual Figure 4 and Fig. 7 could appear at the base of the 3D plots. Therefore, an attempt to illustrate both cases of constant and variable runoff coefficient could also be performed, highlighting an interesting comparison between the two considered cases. However, it is not sure the feasibility.

Reference in the conclusion could be removed. Other general features could be pointed out in the conclusions that could strengthen the paper. An examples is a general conclusion about the influence of the rainfall structure on the dimentionless peak, also associated with the assumption of constant or variable runoff coefficient (see previous point).

The limiting assumptions in the original models, should be mentioned in the text (they are marginally reported in the conclusions). It is not sure that the majority of potential readers of this paper would be familiar with all of them.

A table reporting the parameters associated with the maximum rainfall depth (DDF) of Fig. 1 (at the centre) and Figure 10 (at the bottom) could be useful for the readers.

Although referred to the hillslope scale, a recent paper dealing with the feature considered by the Authors, could be considered for the m/s: Baiamonte, G., Singh, V. P. (2017). "Modelling the probability distribution of peak discharge for infiltrating hillslopes." Water Resour Res, Doi: 10.1002/2016WR020109. With reference to the dimensionless approach: Baiamonte, G., Singh, V.P. (2016). "Analytical Solutions of Kinematic Wave Time of Concentration for Overland Flow under Green-Ampt Infiltration" J Hydrol E – ASCE, 21(3), Doi: 10.1061/(ASCE)HE.1943-5584.0001266, 04015072. Baiamonte, G., Singh, V.P. (2016). "Overland Flow Times of Concentration for Hillslopes of Complex Topography" J Irrig Drain E-ASCE, 142(3), Doi: 10.1061/(ASCE)IR.1943-4774.0000984, 04015059.

Technical corrections

Pag. 3, Line 5. Modify the symbol Tr by typing r subscript

Pag. 3, Line 9. Modify the symbol tr by typing r subscript

Pag. 3, Line 15. Since many depth and time symbols are used, please further define: Rainfall depth, h, …... to the rainfall value of the maximum rainfall depth, hr. Similarly,
for the duration . . ...

Pag. 4, Line 3. Please, correct GIUH

Pag. 4, Eq. (12) can be simplified as (1/(1-exp(.))

Pag 8 Lines 1 -16 not clear. Recommend rewording

Pag 11 Please, insert commas ", corresponding to the scaling exponent of the DDF curves,"

I hope these suggestions can help the Authors to improve their manuscript.

Sincerely

Giorgio Baiamonte
* * *

---

## Referee Comment (RC2) · Anonymous Referee #2 · 15 Sep 2017

The authors present a methodology based on the use of dimensionless rectangular hyetograph and a dimensionless IUH with the aim of predicting the hydrological response of a generic catchment with particular focus on the runoff peak. The method is applied to some events occurred in a small catchment of Northern Italy. My general idea is that I don't like very much the work because seems to me too related on a (useful but..) scholastic hydrology or in such a sense old style hydrology, anyway I admit that this is a little bit my personal and subjective opinion and I think it is not correct evaluate a work on this basis. I think that the topics of this work should be faced with most advanced modeling tools (at least in a research perspective), but I recognize that the presented work could be of a certain interest for those who study and apply

these kind of methodologies (for example on practical applications). In my opinion the paper needs further strong improvement especially on describe motivations, possible problems related to hypothesis, improve case study and application. As final review evaluation I would suggest: Major review Main points: 1)Reasons: the authors should better discuss the reasons of such kind of methodology. For what I understand on one side they propose the methodology as tool to deal with design storms (for example in a projecting process). On the other side refer constantly to internal structure of real event, so in such a way to "reduce" real events in constant intensity hyetographs of variable length. Why doing this? So at the end what is the scope of the method?

2)The hypothesis of constant hyetograph (from line 28 of page 2) is quite strong. This can be motivated in order to simplify the methodology but can lead to "distortion" of results . In the practice in order to produce a project storm, other methodologies are used. For example the Chicago Hyetograph (cited by authors), or individuating a typical duration t1 of rainfall events in a certain area and then nesting an event with duration equal to response time t2 (at the end to consider one of the worst configurations). I think authors should compare their method with something like the latter and discuss the hypothesis and differences in results.

3) The combination of constant hyetograph and a concentrated model (Nash) could lead to some difficulties. When drainage area of catchment increases, the response to intense events can be due to a part of the catchment and the operation of average of rain to obtain a unique hyetograph can lead far from reality. Moreover in a project perspective you should use a multiplicative factor (we can name it kA <1) to reduce the rainfall h derived by DDF, since they generally have punctual meaning; as a consequence kA can become a crucial factor in Qmax estimation when you move from dimensionless to "dimension" values. I think authors should evidence and discus all this issues, since they can have a not negligible effect for such kind of methods (or maybe with same order of magnitude).

4)Initial soil moisture conditions This element seems to be totally negletted, but it impact very much on peak flows and is often a problem during the study of the impact of a certain rainfall storm. So it is possible that using the standard Chicago hyetograph method with AMC3 leads to higher peaks than the proposed method. The issue of contemporaneity of Rainfall with certain T and wet or dry initial condition is a classic problem. I think this should be evidenced and should faced in such way in the presented applications.

5)Application. I do not understand the scope of applying the method to real events. In this case, if I want estimate the Qpeak, supposing to have a calibrated model I should use the rainfall time history, estimate the initial soil moisture and run the model to estimate Qpeak. The analysis done seems to me unuseful (but maybe because it is not clear the scope), what is the reason to build constant hyetograph for different durations picking the magnitude from a real event? You are building un-real rainfall events (and so un real catchment response...) when you already have the truth (..or a truth estimation). If I well understand, in figure 10 a sort of DDF built with hr=80 mm (derived by the events, and which I suppose has a certain return period T*) and n=0.39 (from Mediterranean statistical analysis) is compared with rainfall depth obtained by various n derived for each single event. But what does it mean this comparison? Maybe exist various H(T) > hr (for increasing T) that give same rainfall depths for the different durations but with n=0.39. Maybe the information is only that for those events, for some durations > basin response time (tr) the rainfall depth has a T larger than hr. I suggest: a)on one side better explaining the reasons and motivations of the presented experiment. b)On the other side I would like to see a sort of "project" experiment. So suppose to have the need to estimate the Q for a certain T, considering other methodologies (example Chicago hyetograph ? events with rainfall peaks at the end of hyetograph? Other?), and make a comparison. I think authors should start for the same data (DDF? Rainfall annual maxima on different duration?...) and compare the proposed method with other ones. Moreover I would introduce the effects of Soil Moisture, it is in fact possible that the usage of different AMC introduce more differences than the usage of different methods. Maybe it would be interesting look at the results on different Basins (for example
different drainage area) In practice I would like see a comparison with other methods commonly used in order to evaluate differences in final results and sensitiveness on at least part of common parameters (soil moisture, rainfall reduction area factor kA).

Issues on the paper: 1) Page 1, lines 12-13. What are d* and n?, commonly parameters of equations should not mentioned along the abstract. Moreover in this case no definition is presented. 2) Page 2, lines7-9. Not clear, please explain better. 3) Page 3, line 1. What do you mean with "structure" exponent? 4) Page3, line 9. In general the tr is not fixed and even in case it is consider constant it could vary depending the estimation methodology. Please comment the fact and hypothesis done. 5) Page 4, lines 13-22. If I'm not wrong authors call with same variable tp* two different quantities: the peak of IUH and the peak of UH. 6) Page 4: equation seven, maybe a should be changed in alpha 7) Page 5 line 22: do you refer to equation 19? 8) Page 6 eq 22: what is phir? It is not defined. 9) Page 7 line 20: correct the sentence. It is not clear 10) Page 8 line 3, what is "short-short-short"??? 11) Page 8 line 10-16. It is not clear the sentence/comment about Chicago hyetograph since authors method bases on constant hyetograph while Chicago one not. I suggest to add an example in the results to support this finding. 12) General comment: what do the authors mean with soil abstraction? Do they refer to of th Cn method ?They indicates it with S but from the descriptions it seems they refer to I=0.2*S. 13) Looking at figure 3 and figure 6, apparently the differences are very small in many cases. It is difficult to perceive the effects when you estimate flow (not dimensionless..) 14) Page 9lines 13-20 and figure 9.It is not clear to me if the fact that "highest dimensionless increases with increasing ths S*..." is relevant or not...What does it happens when you go back to"dimension" case? Can you make some comments about the effects of what you found? 15) General comment: it seems to me that in the paper authors sometimes refer to "highest dimensionless runoff peak" and sometimes "maximum dimensionless runoff peaks" (example text on page 9 and figure 9). Con you check the coherence of the terms, variables..etc..along the text? 16) Figure 11 and comments on the text. Are the hyetographs built with "dotted lines" in figure 10? Since the hydrographs are not the

response to the real rainfall event (which is due to particular time sequence of rainfall) , what do they represent? How they vary with different soil moisture condition? What do the reference peak flow (estimated with the described method) represent, and in which sense it is considered a "reference"?

―――――――――――――――――――――――

---

## Author Response (AR1)

The authors would first like to thank the Editor (Fabrizio Fenicia) for taking time out of his schedules to improve the quality of this manuscript. In the below list of detailed answers, the authors have reported each specific comment in bold and the answer is summarized in a section immediately below.

*As already highlighted by both reviewers, a strong limitation of this paper is that it is unclear which aspects of current research it is trying to advance. I understand that the work is about predicting the impact of the rainfall event structure on the peak flow rate. However, it is unclear what has been done before on this subject, and what are the limitations of previous studies which this particular work is attempting to address.*

*Reviewer 2 notes that the paper sounds uninteresting, which in my opinion is due to the fact that it lacks the structure that can make a research work interesting. It is not sufficient to present a methodology and apply it. One needs to motivate it, and to show that it works better than other state of the art approaches. Otherwise, why should one use it?*

*There is considerable work to be done in the introduction, to describe more specifically state of the art approaches, their limitations, and identify the paper objectives. If I read that the first objective of the paper is "to define a structure relationship of the rainfall event in terms of a simple power function" I wonder why this is interesting, if this has already been done, what are the challenges associated with this objective. If you can't defend why pursuing this objective is interesting, then you should look for other objectives.*

The Introduction section has been revised in order to better point both the main focus and the novelty of the presented research. In particular the objectives of the research have been revised and the state of the art has been enlarged by including two more citations (see Baiamonte and Singh, 2017; Alfieri et al., 2008). Finally in the Methodology section, the authors have better specify the meaning of rainfall event structure in order to clarify the novelty of the proposed approach in the framework of the existing literature.

The reviewed version of the text in the Introduction section (pag. 2- lines 6-18) is reported below and put in inverted commas.

"…Baiamonte et al. (2017) investigated the role of the antecedent soil moisture condition in the probability distribution of peak discharge and proposed a modification of the rational method in terms of a-priori modification of the rational runoff coefficients.

In this framework, the present research study takes a different approach by exploring the role of the rainfall event features on the peak flow rate values. Therefore the main objective is to implement a dimensionless analytical framework that can be applied to any study case (i.e. natural catchment) in order to investigate the impact of the rainfall event structure on hydrograph peak. Since the catchment hydrologic response and in particular the hydrograph peak is subjected to a very broad range of climatic, physical, geomorphic and anthropogenic factors, the

focus is posed on catchments where lumped rainfall-runoff model are suitable for deterministic event-based analysis. In the proposed approach, the rainfall event structure is described by investigating the maximum rainfall depths for a given duration $d$ in the range of durations $[d/2; 2d]$ within that specific rainfall event, differently from the statistical analysis of the extreme rainfall events. Other authors (e.g. Alfieri et al., 2008) have previously discussed the accuracy of literature design hyetographs (such as the Chicago hyetograph) for the evaluation of peak discharges during flood event on the contrary the proposed methodology allows to investigate the impact of the above mentioned rainfall event structure on the magnification of the runoff peak neglecting the expected rainfall event features condensed in the Depth-Duration-Frequency (DDF) curves.

The first specific objective is to define a structure relationship of the rainfall event able to describe the sample space of the rainfall event structures by means of a simple power function. The second specific objective is to implement a dimensionless approach that allows to generalize the assessment of the hydrograph peak irrespective of the specific catchment characteristic (such as the hydrologic response time, the variability of the infiltration process, etc.) thus focusing on the impact of the rainfall event structure.

Finally a specific catchment application is discussed in order to point out the dimensionless procedure implications and to provide some numerical examples of the rainfall structures with respect to observed rainfall events; furthermore their effects on the hydrograph peak are examined."

*The real data case study is very limited. I expected that a paper that explores "peak flow rate values, which are subject to a very broad range of climatic, physical, geomorphic and anthropogenic factors" tests the proposed methodology on a broad range of climatic, physical, geomorphic and anthropogenic factors. Instead, the method is applied to one catchment, and I did not find a clear comparison between predictions and observations of peak flow rate values. I would expect a case study that compares predictions and observations of peak flow rate values in catchment with contrasting responses, to show that the method is able to capture the differences in catchment behaviour.*

The aim of the present research is to develop an analytical framework in order to assess the impact of the rainfall event structure on the magnification of the peak flow rate. The catchment application is designed in order to support the understanding of the proposed dimensionless methodology rather than to numerical validate it. Indeed the verification of the hydrologic model suitability in predicting the hydrograph peak at given return periods or for observed rainfall events is out of the scope of the present research. In particular the authors, in the answer to comment SC1 of Ref#1, have reported the analysis of an observed rainfall event for which flow rate data are available. The results of this analysis are limited to a single implication: the observed dimensionless hydrograph peak represents only one of the outcomes in the sample space of the pair $n$-structure/ peak that in the specific case is close to the maximum one; however other conditions (more severe or less critical) could be expected for that internal structure of the event.

Furthermore, the dimensionless procedure itself allows generalizing the results to different case studies characterized by a very broad range of climatic, physical, geomorphic and anthropogenic factors; on the contrary the application to several catchments is not useful and does not add any additional results.

Finally the authors have reworded the objectives and the preface of the methodology section in order to improve the understanding of the objectives of the research, to clarify the analytical framework and to better specify the catchment application (see also the answers to the comments C1 and C5 of Ref#2).

*When reading the methodology I found it difficult to understand where it is going to lead. I recommend to state first why you do something, and then how you do it. This could contribute to clarify the linkage between the given sequence of steps.*

The preface of the Methodology section has been completely revised in order to improve its readability and understanding (see also the answer to comment SC5 of Ref#1). Furthermore in the subsection 2.1, the dimensionless approach is better specified by clarifying the role of the reference values of the rainfall depth and duration in the reading of the dimensionless results. Regarding the subsections 2.2 and 2.3, few specifications have been included regarding the shapes of the IUH and the hyetograph in order to detail the methodological framework. Finally the subsection 2.3.2 has been revised in order to better point out the role of the variable runoff coefficient (see also the answer to comment C4 Ref#2).

*As a last minor comment, I found the wording "rainfall event structure" poorly defined. I was surprised that after speaking of rainfall event structure, a constant hyetograph was used.*

The authors have revised the subsection 2.1. "The dimensionless form of the rainfall structure relation" by including a precise definition of the rainfall event structure and its implications in the characterization of a rainfall event (see also the response to comment SC2 Ref#1 and C1 Ref#2).

As for the use of a constant hyetograph in the dimensionless approach, it has to be noticed that it is functional to the description of the rainfall event structure that is represented by means of a simple power-function. Since the analysis is carried out according to a dimensionless framework the use of a specific shape of the hyetograph (in this case constant for the above mentioned reason) does not affect the results that could be similarly carried out even for other hyetograph shapes (see also the response to comment C2 of Ref#2).

**Section 2.a Point-by-point reply to the Interactive comment of the Referee #1 (Giorgio Baiamonte) on "A dimensionless approach for the runoff peak assessment: effects of the rainfall event structure" by Ilaria Gnecco et al.**
The authors would first like to thank the Referee#1 (Giorgio Baiamonte) for taking time out of his schedules to improve the quality of this manuscript. In the below list of detailed answers, the authors have reported each specific comment in bold and the answer is summarized in a section immediately below.

**Ref.#1 Specific Comment SC1:**
*The only concern about this paper is the lacking of a comparison of the analytical approach to the hydrological response with some available analytical model previously published in order to deepen its degree of applicability. Otherwise, the main findings could be compared with experimental or numerical data that are available in the literature. This should not be too much time consuming to achieve and it will significantly strengthen the paper. The application for the Bisagno catchment could not be considering as a test of the suggested solution. Therefore, my recommendation is to try to achieve such a task.*
 *Answer SC1*

The authors intend to confirm that the proposed analytical approach is applicable to all the lumped model proposed in the literature (e.g. Chow et al., 1988): by selecting other IUH forms/derivation and/or different soil abstraction model, the main findings (i.e. the relationships between the *n* structure exponent and the maximum dimensionless hydrograph peak for a given dimensionless duration) should be numerically different but substantially comparable. In the manuscript the contour plots illustrated in Figs 7 and 12 (that are referred to the numerical example and to the catchment application, respectively) are numerically different since both the shape parameter of the gamma function IUH (i.e. 3 and 3.4, respectively) and the dimensionless soil abstraction (i.e. 0.25 and 0.5, respectively) are different. However the main findings and implications derived by the two contour plots are the same. The use of the Nash IUH allows to derive analytically the relationship between the maximum dimensionless peak and the *n* structure for a given dimensionless duration; similarly analytical derivation could be carried out for simple synthetic IUHs. Furthermore, for experimentally derived IUH even if the analytical solution of the problem is not feasible, the proposed methodology can be performed to calculate numerically the maximum dimensionless peak for given *n* structure and dimensionless duration values.

The authors want to underline that the main objective of the present research is to investigate the relationship between the hydrograph peak and the rainfall event structure rather than predict the hydrologic response of a catchment at a given rainfall event. Therefore a 'classical' verification/validation of the proposed approach with experimental or numerical data is not meaningful and does not contribute to properly assess its suitability. Indeed, in order to point out the internal structure of the rainfall events, the observed rainfall events are represented by means of simple constant hyetograph characterized by specific $n$ structure exponent and duration; then, in order to assess the impact on the hydrograph peak, all the possible rainfall structures in the range [0.2; 0.8] and duration in the range $[t_r/2; 2t_r]$ have been analysed (see also Answer SC2). By considering that any observed rainfall event shows (for each duration) a specific $n$ structure value, that represents only one of the possible outcomes in the sample space of the rainfall structure values, the corresponding observed hydrograph peak should be one of the possible outcomes that could not be necessarily the most sever one (i.e. the maximum expected value).

In spite of the previous consideration, in order to clarify and support this answer, the authors have completed the application of the Bisagno catchment at La Presa station with the analysis of one observed rainfall-runoff event for which flow rate data are available (since published in Gabellani et al., 2008). The analysis involved the rainfall-runoff event occurred the 25[th] of November 2002. This event is characterized by the reference rainfall depth of 46.9 mm and by the $n$ structure value of 0.63, evaluated for the duration equal to the Bisagno – La Presa catchment reference time (i.e. 0.85 h). The reference depth and the $n$ structure value of the observed rainfall event are summarized in the Table 2 together with the main characteristics of the corresponding generated events. The excess depths are evaluated for each duration by assuming the soil abstraction equal to 41 mm accordingly with the $CN_{II}$ value estimated for the catchment (Bocchiola and Rosso, 2009). The hyetograph and the corresponding hydrograph observed at the Bisagno – La Presa catchment together with the reference value of the runoff peak flow are reported in the top graph of Fig.13. The rainfall structure curve and the corresponding Depth-Duration-Frequency curve evaluated for the reference time are plotted in the central graph of Fig.13 while the contour plot of the dimensionless runoff peak in the bottom one. Note that, in the bottom graph, the maximum dimensionless runoff peak curve (bold line) and the observed dimensionless hydrograph peaks (red-filled stars) are also reported. The observed dimensionless hydrograph peaks is close to the maximum one however other conditions (more severe or less critical) could be expected for that internal structure of the event.

In this case, even if the authors have compared the results with experimental data as suggested by the Reviewer, they do not consider that the Fig. 13new and Table 2 should necessarily be included in the manuscript.

Finally, in order to improve the understanding of the proposed approach, the authors have better specified the dimensionless framework illustrated in the Section 2.2 (lines 1-9 of page 4). The reviewed version of the text is reported below and put in inverted commas.

"The use of the Nash IUH allows to define an analytical framework to assess the relationship between the maximum dimensionless peak and the $n$ structure exponent for a given dimensionless duration and similar analytical derivation can be carried out for simple synthetic IUHs.

….

The proposed dimensionless approach is based on the use of the IUH scale parameter as the reference time of the hydrologic response (i.e. $t_r = \alpha k$). Using the first order moment in the dimensionless procedure, the proposed approach can be applied to any IUH form even if, for experimentally derived IUH, the analytical solution of the problem is not feasible. "

**Table 2: Reference depth and *n* structure value of the observed rainfall event for the Bisagno – La Presa catchment and main characteristics of the corresponding generated ones.**

| Rainfall event [dd/mm/aaaa] | Reference depth [h] | n structure [-] | Duration [h] | Depth [mm] | Excess depth $h_e$ [mm] | Excess intensity $i_e$ [mm/h] |
|---|---|---|---|---|---|---|
| 25/11/2002 | 46.9 | 0.63 | 0.425 | 30.2 | 12.8 | 30.2 |
| | | | 0.85 | 46.9 | 25.0 | 29.5 |
| | | | 1.275 | 60.7 | 36.2 | 28.4 |
| | | | 1.7 | 72.8 | 46.6 | 27.4 |

[Figure]

**Figure 13: Hyetograph and the corresponding hydrograph observed at Bisagno – La Presa catchment together with the reference value of the hydrograph peak flow (at the top); rainfall structure curve and the corresponding Depth-Duration-Frequency curve evaluated for the reference time (at the centre); contour plot of the dimensionless runoff peak (at the bottom). In the bottom graph, the maximum dimensionless runoff peak curve (bold line) and the observed dimensionless hydrograph peaks (red-filled stars) are also reported.**

**Ref.#1 Specific Comment SC2:**

*It is not clear to me how the rainfall structure exponent in the example of Figure 1 (at the bottom) and in the application for the Bisagno catchment at La Presa station (Figure 10 at the centre) is determined. Although based on previous studies, n exponent determination vs rainfall duration should be described. How does n qualitatively influence the rainfall structure? I checked Figure 1 at the centre, is the estimated one a simple power law? Please, add parameters (the same for Fig. 10).*

*Answer SC2*

The authors agree that the estimation of the *n* structure exponent values associated to a given rainfall event is not clearly reported in the manuscript, even if the aim of Figure 1 is to provide a practical example of the internal structure with respect to an observed rainfall event.

In the proposed approach, the authors assumed that the maximum rainfall depth for a given duration observed in each rainfall event can be described in terms of a power function similarly to the DDF curve:

$$h(d) = a'd^n \tag{1}$$

where h [L] is the maximum rainfall depth, $a'$ [LT$^{-n}$] and *n* [-] are respectively the coefficient and the structure exponent of the power function for a given duration, $d$ [T]. For each duration $d_i$, the corresponding power function parameters (i.e. $a'$ and *n*) are estimated based on the maximum rainfall depth values observed in the range of duration $[d_i/2; 2d_i]$ by means of a simple linear regression analysis. Based on such assumptions, a given rainfall event that is characterized by a specific *n* structure exponent at a given duration is only one of the possible outcomes in the sample space of the rainfall structures. In other words, the structure exponent *n* allows describing the rainfall event based on a simple rectangular hyetograph thus representing the rainfall event structure at a given duration. Indeed assuming a rainfall depth in a given duration as a reference/equivalent rainfall value (named respectively as $h_r$ and $t_r$), the rainfall event structure may be significantly different with *n* structure exponent values that can mathematically range between 0 and 1. The two extreme values represent un-realistic events characterized by opposite event structure: when the structure exponent *n* tends to zero the structure of the rainfall event is comparable to a Dirac impulse while it is comparable to a constant intensity rainfall for *n* close to one.

For example, the *n* structure exponent is evaluated on hourly basis with respect to four observed rainfall events as illustrated in Figures 1 and 10 (black dots). According with the definition of the rainfall event structure, above mentioned, the "estimated" curve reported in Fig. 1 (at the centre) is not a simple power law but it is the ensemble of all the specific regression curves estimated for each duration, $d_i$, in the range of duration $[d_i/2; 2d_i]$; thus the authors have decided to remove that curve to avoid misinterpretations (see Fig. 1rev and Answer SC6).

On the other hand, in order to improve the readability of the manuscript the authors have included the $a'$ and $n$ power function parameters of the rainfall structure curve in each graph reported at the bottom of Figure 10 (see Fig. 10rev and Answer SC6). It has to be noticed that the $a'$ values can be evaluated only with respect to a given reference rainfall depth and consequently a given the reference time, $t_r$. It follows that the $a'$ values reported in Fig. 10rev are valid for the Bisagno – La Presa catchment application characterized by a reference rainfall depth of 80 mm and a reference time of 0.85 h.

Finally, the authors have largely revised the subsection 2.1. in order to clearly illustrate how the structure exponent $n$ is calculated.

[Figure]

**Figure 1rev: Rainfall event structure: the observed rainfall depth (at the top), the observed maximum rainfall depths (at the centre), and the corresponding rainfall structure exponent (at the bottom) are reported.**

[Figure]

**Figure 10rev: Rainfall event structure of three events observed in Genoa (IT): the observed rainfall depths (at the top) and the estimated rainfall structure exponents (at the centre) are reported. At the bottom, the rainfall structure and Depth-Duration-Frequency curves, evaluated for the reference time of the Bisagno – La Presa catchment, are reported.**

**Ref.#1 Specific Comment SC3:**

*Figure 4 and 7 are very effective. However, to show the effect of the rainfall structure parameter, it could be useful also plotting the dimensionless hydrograph peak vs rainfall structure exponent, for both constant and time-varying psi. Moreover, a 3D figure could better evidence the influence of the rainfall structure on the dimensionless peak discharge and the saddle area. The actual Figure 4 and Fig. 7 could appear at the base of the 3D plots. Therefore, an attempt to illustrate both cases of constant and variable runoff coefficient could also be performed, highlighting an interesting comparison between the two considered cases. However, it is not sure the feasibility.*

*Answer SC3*

As suggested by the Reviewer, the authors have modified the Figs. 4 and 7 by coupling the contour plot of the dimensionless hydrograph peak to the 3D graph in order to better highlight its behaviour as a function of the rainfall structure exponent and the dimensionless rainfall duration thus posing particular attention to the saddle area. The revised Figures reported below (see Figs. 4rev and 7rev) support the understanding of the main results of the 3.1 and 3.2 sections. In particular, looking at Figs. 4 rev and 7 rev, it emerges that in case of variable runoff coefficient, the range of variation of the dimensionless hydrograph peak is wider with respect to the constant runoff coefficient. In particular, when the dimensionless rainfall duration increases ($d^* > 1$), the dimensionless hydrograph peak is higher in case of variable runoff coefficient than in constant case while the opposite occurs for short durations ($d^* < 1$) (the surface is steeper in Fig. 7rev with respect to Fig. 4rev).

On the other hand, due to the complexity of the Figures (both in the actual and revised versions) the authors have proposed the comparison between results with respect to the constant and variable runoff coefficient cases in Fig. 9, thus focusing on the maximum hydrograph peak associated with a specific rainfall structure in terms of $n$ value and dimensionless time-to-peak (i.e. dimensionless rainfall duration, see also Figs. 5 and 8). Note that for a given dimensionless time-to-peak (see Eq. 9 and Fig. 9) or dimensionless duration (see Figs. 4rev. and 7rev.), the maximum hydrograph peak is associated with $n$ structure exponent that decreases moving from the constant runoff coefficient case to the variable ones, indeed the rate of change in the runoff production ascribable to the variable runoff coefficient is predominant with respect to the one due to the rainfall duration increase.

[Figure]

**Figure 4rev: 3D mesh plot (at the top) and contour plot (at the bottom) of the dimensionless hydrograph peak as a function of the rainfall structure exponent and the dimensionless rainfall duration in case of constant runoff coefficient. The maximum dimensionless runoff peak curve is also reported (bold line).**

[Figure]

**Figure 7rev: 3D mesh plot (at the top) and contour plot (at the bottom) of the dimensionless hydrograph peak as a function of the rainfall structure exponent and the dimensionless rainfall duration in case of variable runoff coefficient. The maximum dimensionless runoff peak curve is also reported (bold line).**

**Ref.#1 Specific Comment SC4:**

*Reference in the conclusion could be removed. Other general features could be pointed out in the conclusions that could strengthen the paper. An example is a general conclusion about the influence of the rainfall structure on the dimensionless peak, also associated with the assumption of constant or variable runoff coefficient (see previous point).*

*Answer SC4*

The authors have removed the reference in the conclusion section and have better pointed out the impact of the rainfall event structure as suggested by the reviewer. The conclusion section have been largely revised as follows:

"The proposed analytical dimensionless approach allows investigating the impact of the rainfall event structure on the hydrograph peak. At this aim a methodology to describe the rainfall event structure is proposed based on the similarity with the Depth-Duration-Frequency (DDF) curves. The rainfall input consists in a constant hyetograph where all the possible outcomes in the sample space of the rainfall structures can be condensed through the *n* structure exponent. The rainfall-runoff processes are modelled using the Soil Conservation Service (SCS) method for soil abstractions and the Instantaneous Unit Hydrograph (IUH) theory. In the present paper the two-parameter gamma distribution is adopted as IUH form; however the analysis can be repeated using other synthetic IUH forms obtaining similar results.

The proposed dimensionless approach allows defining an analytical framework that can be applied to any study case for which the model assumptions are valid; the site-specific characteristics (such as the morphologic and climatic characteristics of the catchment) are no more relevant being included within the parameters of the dimensionless procedure (i.e. $h_r(T_r)$ and $t_r$) thus allowing to figure out the implication on the hydrograph peak irrespective of the absolute value of the rainfall depth (i.e. the corresponding return period).A set of analytical expressions has been derived to provide the estimation of the maximum peak with respect to a given *n* structure exponent. Results reveal the impact of the rainfall event structure on the runoff peak thus pointing out the following features:

- the curve of the maximum values of the runoff peak reveals a local minimum point (saddle point);
- different combinations of *n* structure exponent and rainfall duration may determine similar conditions in terms of runoff peak;
- analogous behaviour of the maximum dimensionless runoff peak curve is observed for different runoff coefficients although wider range of variation are observed with increasing soil abstraction values.

Referring to the Bisagno – La Presa catchment application ($h_r$= 80mm; $t_r$= 0.85 h and $S_*$= 0.5), the saddle point of the runoff peaks is located in the neighbourhood of *n* value equal to 0.3 and rainfall duration corresponding to the reference time ($d_*$ =1). Further, it emerge that the maximum runoff peak value, corresponding to the scaling exponent of the DDF curve, is comparable to the less critical one (saddle point)."

**Ref.#1 Specific Comment SC5:**

*The limiting assumptions in the original models, should be mentioned in the text (they are marginally reported in the conclusions). It is not sure that the majority of potential readers of this paper would be familiar with all of them.*

*Answer SC5*

    The authors have revised the preface of the methodology section (lines 20-23 of page 2) in order to anticipate to readers the assumptions of the model and consequently to improve the readability of the results and conclusions sections. The reviewed version of the text is reported below and put in inverted commas.

    "The rainfall event is then described as simple hyetographs of a given durations; this simplification is consistent with the use of deterministic lumped models based on the linear system theory (e.g. Bras, 1990). The proposed approach is therefore valid within a framework that assumes that the watershed is a linear causative and time invariant system, where only the rainfall excess produces runoff. In detail, the rainfall-runoff processes are modelled using the Soil Conservation Service (SCS) method for soil abstractions and the Instantaneous Unit Hydrograph (IUH) theory. Consistently with the assumptions of the UH theory, the proposed approach is strictly valid when the following conditions are maintained: the known excess rainfall and the uniform distribution of the rainfall over the whole catchment area."

**Ref.#1 Specific Comment SC6:**

*A table reporting the parameters associated with the maximum rainfall depth (DDF) of*
*Fig. 1 (at the centre) and Figure 10 (at the bottom) could be useful for the readers.*

*Answer SC6*

    Referring to the rainfall event presented in the catchment application, the authors have calculated the $a'$ and $n$ power function parameters of the rainfall structure curve together with the $a$ and $b$ parameters of the DDF curve for the reference time of the Bisagno – La Presa catchment. All the parameters have been added in the legend of each graph at the bottom of Figure 10rev (see Fig. 10rev and Answer SC2).

    The authors have not included the parameters of the "estimated" curve in the Fig. 1 (at the centre) since that curve is not the regression line for all the durations but it is the ensemble of all the specific regression curves estimated for each duration $d_i$ in the range of duration $[d_i/2; 2d_i]$; furthermore the authors have removed the "estimated" curve in Fig. 1 (at the centre) in order to avoid a misleading interpretation (see Fig. 1rev and Answer SC2).

**Ref.#1 Specific Comment SC7:**

*Although referred to the hillslope scale, a recent paper dealing with the feature considered*
*by the Authors, could be considered for the m/s: Baiamonte, G., Singh, V. P. (2017). "Modelling the probability distribution of peak discharge for infiltrating hillslopes." Water Resour Res, Doi: 10.1002/2016WR020109.*

*With reference to the dimensionless approach: Baiamonte, G., Singh, V.P. (2016). "Analytical Solutions of Kinematic Wave Time of Concentration for Overland Flow under Green-Ampt Infiltration"J Hydrol E – ASCE, 21(3), Doi: 10.1061/(ASCE)HE.1943-5584.0001266, 04015072.*

*Baiamonte, G., Singh, V.P. (2016). "Overland Flow Times of Concentration for Hillslopes of Complex Topography" J Irrig Drain E-ASCE, 142(3), Doi: 10.1061/(ASCE)IR.1943-4774.0000984, 04015059.*

*Answer SC7*

The authors have read the papers suggested by the reviewer and included the reference, Baiamonte et al. (2017), in the introduction section at line 6 of page 2. The reviewed version of the text is reported below and put in inverted commas.

"Baiamonte et al. (2017) investigated the role of the antecedent soil moisture condition in the probability distribution of peak discharge and proposed a modification of the rational method in terms of a-priori modification of the rational runoff coefficients".

**Ref.#1 Technical corrections TC:**

*Pag. 3, Line 5. Modify the symbol Tr by typing r subscript*

*Pag. 3, Line 9. Modify the symbol tr by typing r subscript*

*Pag. 3, Line 15. Since many depth and time symbols are used, please further define:*

*Rainfall depth, h, ..... to the rainfall value of the maximum rainfall depth, hr. Similarly, for the duration .....*

*Pag. 4, Line 3. Please, correct GIUH*

*Pag. 4, Eq. (12) can be simplified as (1/(1-exp(.))*

*Pag 8 Lines 1 -16 not clear. Recommend rewording*

*Pag 11 Please, insert commas ", corresponding to the scaling exponent of the DDF*

*curves,"*

*Answer TC*

The sentences, symbols and equations have been revised according to the reviewer suggestion.

In particular, the Lines 1-16 of page 8 reporting the discussion of the results illustrated in Figure 5 have been reworded as follows:

"In Figure 5, the maximum dimensionless runoff peak and the corresponding rainfall structure exponent are plotted vs. the dimensionless time-to-peak. Further, the dimensionless IUH and the corresponding dimensionless UH for $d_*=1.0$ are reported as an example. The reference line (indicated as short-short-short dashed grey line in Fig. 5) illustrates the lower control line corresponding to the rainfall duration infinitesimally small. Note that the rainfall structure exponent that maximizes the runoff peak for a given duration can be simply derived as a function of the dimensionless time-to-peak (see Eq. 20). The maximum dimensionless runoff peak curve tends to one for long

dimensionless rainfall duration ($d_*>3$) when the corresponding $n$ structure exponent tends to one (see Eq. 18): for high-values of $n$ structure, the critical conditions occur for long durations that correspond to paroxysmal events for which the rainfall intensity remains fairly constant. The local minimum of the maximum dimensionless runoff peak curve (see Fig.5) occurs at $t_{p*}$ of 1.29 corresponding to $n$ structure value of 0.31 and $d_*$ of 1, thus pointing out that the less critical runoff peak occurs at $n$ structure exponent values corresponding to the ones typically derived by the statistical analysis of the annual maximum rainfall depth series in Mediterranean climate. Furthermore, it can be observed that different rainfall event conditions (i.e. rainfall structure exponent $n$ and duration $d_*$) in the neighborhood of the local minimum point could determine comparable effects in term of the runoff peak value."

Finally, the Eq. (12) has been simplified as follows:

$$f\left(t_{p*}\right)=f\left(t_{p*}-d_*\right)\ \rightarrow\ t_{p*}=d_*\frac{e^{\frac{\alpha d_*}{\alpha-1}}}{e^{\frac{\alpha d_*}{\alpha-1}}-1}=\ d_*\frac{1}{1-e^{-\frac{\alpha d_*}{\alpha-1}}} \tag{12}$$

The authors would first like thank the Referee#2 for taking time out of his schedules to improve the quality of this manuscript. In the below list of detailed answers, the authors have reported each specific comment in bold and the answer is summarized in a section immediately below.

**Ref.#2 Comment C1:**

*Reasons: the authors should better discuss the reasons of such kind of methodology. For what I understand on one side they propose the methodology as tool to deal with design storms (for example in a projecting process). On the other side refer constantly to internal structure of real event, so in such a way to "reduce" real events in constant intensity hyetographs of variable length. Why doing this? So at the end what is the scope of the method?*

*Answer C1*

> The main objective of the manuscript is to investigate the impact of the internal rainfall event structure on the hydrograph peak. The original contribution of the paper consists, firstly, in the methodology proposed to describe the internal structure of the rainfall event based on the similarity with the DDF curves. The internal structure of the rainfall event is described by means of the $n$ structure exponent (as well as the coefficient $a'$) that is assumed varying across the rainfall event; in particular each $n$ structure exponent is assumed as representative of the rainfall event structure in the range of duration $[d_i/2; 2d_i]$ from which it is derived. Based on such assumptions, the observed rainfall event that is characterized by a specific $n$ structure exponent is only one of the possible outcomes in the sample space of the rainfall structures. In other words, the structure exponent $n$ at a given duration, $d_i$, allows describing the rainfall event based on a simple rectangular hyetograph thus representing the rainfall event structure in the range of duration $[d_i/2; 2d_i]$. Indeed assuming a rainfall depth in a given duration as a reference/equivalent rainfall event (named respectively as $h_r$ and $t_r$), the rainfall event structure may be significantly different: when the structure exponent $n$ tends to zero the internal structure of the rainfall event is comparable to a Dirac impulse while it is comparable to a constant intensity rainfall for $n$ close to one. The second original contribution consists in the dimensionless approach that allows defining an analytical framework that can be applied to any study case (i.e. natural catchment) for which the model assumptions are valid (i.e. linear causative and time invariant system). The reference values $h_r$ and $t_r$ are directly linked to the climatic and morphologic characteristics of the specific catchment, therefore the dimensionless approach based on the $h_r$ and $t_r$ values allows to generalize the results irrespective of the specific catchment characteristic (such as the return period associated to the reference rainfall event) thus focusing on the impact of the structure exponent $n$ (i.e. the rainfall event structure) on the hydrograph peak.

In order to improve the readability and understanding of the proposed methodology, the authors have largely revised the Section 2.1 in order to clearly illustrate how the structure exponent *n* is calculated and to better point out the influence of the internal structure on the characterization of a rainfall event.

**Ref.#2 Comment C2:**

*The hypothesis of constant hyetograph (from line 28 of page 2) is quite strong. This can be motivated in order to simplify the methodology but can lead to "distortion" of results. In the practice in order to produce a project storm, other methodologies are used. For example the Chicago Hyetograph (cited by authors), or individuating a typical duration t1 of rainfall events in a certain area and then nesting an event with duration equal to response time t2 (at the end to consider one of the worst configurations). I think authors should compare their method with something like the latter and discuss the hypothesis and differences in results.*

*Answer C2*

The hypothesis of constant hyetograph is not motivated in order to simplify the methodology as previously discussed (see also the answer to the comment C1). In order to describe the internal structure, the rainfall event is represented by means of a power function where the parameters are not constant as in the DDF curves but depend on duration. Based on such approach, the rainfall event structure at a given duration is represented throughout the *n* structure exponent, it follows that the rainfall event can be described by a simple rectangular hyetograph. In has to be noticed that the constant hyetograph derived by a given *n* structure is assumed valid in the same range of duration $[d_i/2; 2d_i]$ from which it is derived. In order to point out the rainfall event structure that causes the maximum hydrograph peak and, in general, how the rainfall event structure affect the hydrograph peak, the hydrologic response of a catchment has been analytically derived using a deterministic lumped model to describe the rainfall-runoff process and considering the sample space of the rainfall event structures by varying the *n* structure exponent in the range [0.2; 0.8].

On the other hands, the authors state that the aim of this work is not to assess the accuracy of literature design hyetographs (such as the Chicago hyetograph) for the evaluation of peak discharges during flood event, other authors have previously discussed that (e.g. Alfieri et al., 2008); the main goal is to assess the impact of the rainfall event structure on the magnification of the runoff peak. Other forms of hyetographs could produce hydrograph peak estimates that are consistently different: we agree with the Reviewer that the rectangular hyetograph tends to underestimate the peak flows with respect to the Chicago hyetograph, however the proposed methodology is not addressed to the robust estimation of the peak flows but it is addressed to enhance the impact on the peak flow rate of the rainfall structure. Finally, the rectangular hyetograph allows deriving analytically the relationship between the maximum peak and the *n* structure value for a given duration; however the proposed approach could be implemented with different hyetograph shape (even if numerical calculation is required instead of the proposed analytical derivation).

In order to avoid a misleading interpretation of the presented analytical framework, the reference to the Chicago hyetograph (pag.8, lines 10-14) has been removed from the text.

Furthermore the authors have included a new paragraph in the Section 2.3, in order to explain the meaning of the constant hyetograph in the presented approach. The reviewed version of the text is reported below and put in inverted commas.

"Note that the hypothesis of rectangular hyetograph is not motivated in order to simplify the methodology but in order to describe the rainfall event structure. Based on such approach, the rainfall event structure at a given duration is represented throughout the $n$ structure exponent, it follows that the rainfall event is described by a simple rectangular hyetograph. It has to be noticed that the constant hyetograph derived by a given $n$ structure is assumed valid in the same range of duration from which it is derived $[d_i/2; 2d_i]$."

**Ref.#2 Comment C3:**

*The combination of constant hyetograph and a concentrated model (Nash) could lead to some difficulties. When drainage area of catchment increases, the response to intense events can be due to a part of the catchment and the operation of average of rain to obtain a unique hyetograph can lead far from reality. Moreover in a project perspective you should use a multiplicative factor (we can name it kA <1) to reduce the rainfall h derived by DDF, since they generally have punctual meaning; as a consequence kA can become a crucial factor in Qmax estimation when you move from dimensionless to "dimension" values. I think authors should evidence and discus all this issues, since they can have a not negligible effect for such kind of methods (o maybe with same order of magnitude).*

*Answer C3*

The proposed methodological approach involves a deterministic lumped model based on the linear system theory (UH theory) for which a watershed's runoff response is linear and time-invariant and the excess rainfall occurs uniformly over the watershed. The authors are well conscious that the areal distribution of precipitation affects the hydrologic response thus the hydrograph peak, however it is out of the scope of the present research study. Indeed the proposed methodology should not be used to predict the hydrologic response of a given catchment to a real rainfall event where it is crucial to count for the areal distribution of precipitation especially for large catchment area. Furthermore, consistently with the assumptions of the UH theory, the proposed approach is strictly valid when the following conditions are maintained: the linearity and time invariance of the response function, the known excess rainfall, and the uniform distribution of the rainfall over the whole catchment area.

The authors have revised the preface of the methodology section (lines 20-23 of page 2) in order to clarify to the readers the assumptions of the model and consequently to improve the understanding of the results and conclusions sections. The reviewed version of the text is reported below and put in inverted commas.

"The rainfall event is then described as simple hyetographs of a given durations; this simplification is consistent with the use of deterministic lumped models based on the linear system theory (e.g. Bras, 1990). The proposed approach is therefore valid within a framework that assumes that the watershed is a linear causative and time invariant system, where only the rainfall excess produces runoff. In detail, the rainfall-runoff processes are modelled using the Soil Conservation Service (SCS) method for soil abstractions and the Instantaneous Unit Hydrograph (IUH) theory. Consistently with the assumptions of the UH theory, the proposed approach is strictly valid when the following conditions are maintained: the known excess rainfall and the uniform distribution of the rainfall over the whole catchment area."

**Ref.#2 Comment C4:**

*Initial soil moisture conditions This element seems to be totally negletted, but it impact very much on peak flows and is often a problem during the study of the impact of a certain rainfall storm. So it is possible that using the standard Chicago hyetograph method with AMC3 leads to higher peaks than the proposed method. The issue of contemporaneity of Rainfall with certain T and wet or dry initial condition is a classic problem. I think this should be evidenced and should faced in such way in the presented applications.*

*Answer C4*

The authors agree that the initial soil moisture conditions as well as the variability of the infiltration process across the rainfall event significantly affect the hydrological response of the catchment. Indeed the authors include the influence of the infiltration process occurring at each rainfall event by means of a variable runoff coefficient that is estimated based on the SCS method. In particular, the excess rainfall depth is evaluated as a function of the total rainfall depth, $h$ and the soil abstraction parameter, $S$ (see Eq. 21).

According to the dimensionless approach proposed in the present paper, the dimensionless soil abstraction $S_*$ is defined as the ratio of $S$ to the reference rainfall depth, $h_r$. Therefore different initial moisture conditions (i.e. $CN_I$ or $CN_{III}$ values and consequently the computing of the $S(CN_I)$ or $S(CN_{III})$ values) are included and analyzed in the proposed methodology by considering different $S_*$ associated to the same reference rainfall depth. An attempt to show the impact of different soil moisture conditions is provided in Fig. 9. In order to point out the influence of different variable runoff coefficients (i.e. initial moisture conditions) Fig. 9 illustrates the maximum dimensionless hydrograph peak (see the top graph) and the corresponding rainfall structure exponent (see the centre graph) vs. the dimensionless time-to peak with respect to $S_*$ values of 0.25 and 0.67.

The authors have partially revised the Sections 2.3.2 and 3.2 in order to clearly mention the impact of the initial soil moisture conditions.

Furthermore, to better point out that the different initial soil moisture conditions are taken into account in the proposed approach, the authors have here included Figs. 11new and 12new where different initial moisture conditions (i.e. $CN$ values) are considered in the catchment application. Looking at Fig. 11new, different $CN$ values affect the excess rainfall intensity thus the hydrograph peak and the reference peak flow values that increase with increasing $CN$, as expected. By comparing the graphs reported in Fig.12new, it emerges that the range of variation of the dimensionless hydrograph peak is wider when the $S_*$ value increases, such behaviour is due to the rate of change in the runoff production with respect to the rainfall duration: with increasing the rainfall volume the relevance of runoff with respect to the soil abstraction rises. It has to be noticed that in spite of such wider range of variation of the dimensionless hydrograph peak, the increasing of $S_*$ value corresponds, in dimensional term, to the decreasing of the $CN$ value (assuming constant the reference rainfall depth), it follows that the reference peak flow value decreases.

The author wouldn't like to include the Fig. 11new and 12new in the text since they are very full of information thus not really supporting the paper readability, on the other hands, they have revised Figs. 11 and 12 adding the $CN$ and $S$ values in the graph and reworded the comment to Fig. 9 (lines 11-21 pag.9) in order improve the understanding of the variable runoff coefficient case.

Although the authors are conscious that the effect of initial moisture conditions on the hydrologic response of a catchment is a classic problem that affects the iso-frequency hypothesis between rainfall and runoff and deeply debated in the literature (see e.g. De Michele and Salvadori, 2002); the authors want to point out again that the main objective of the paper is to assess the impact of the rainfall event structure on the peak flow rate by means of a deterministic lumped model based on the linear system theory. The evaluation of the runoff peak associated to an observed rainfall event is out of the scope of the present approach.

[Figure]

**Figure 11new: The excess rainfall hyetographs, the corresponding hydrographs and the reference value of the hydrograph peak flow for the Bisagno – La Presa catchment evaluated for three rainfall structure exponents and three soil abstraction ($CN_I$, $CN_{II}$ and $CN_{III}$). Note that each graph includes four rainfall durations (i.e. 0.5, 1.0, 1.5, and 2.0 times the reference time).**

[Figure]

**Figure 12new: Contour plot of the dimensionless runoff peak evaluated for the Bisagno – La Presa catchment for three different soil abstraction ( $S_* = 0.2$, 0.5 and 1.2). In each graph, the maximum dimensionless runoff peak curve (bold line) is also reported together with the dimensionless hydrograph peaks (grey-filled stars) for the selected rainfall structure exponents ($n$ = 0.55, 0.62, 0.71) and durations ($d_*$ = 0.5, 1.0, 1.5, and 2.0).**

[Figure]

**Figure 11rev: The excess rainfall hyetographs, the corresponding hydrographs and the reference value of the hydrograph peak flow for the Bisagno – La Presa catchment evaluated for three rainfall structure exponents. Note that each graph includes four rainfall durations (i.e. 0.5, 1.0, 1.5, and 2.0 times the reference time).**

[Figure]

**Figure 12rev: Contour plot of the dimensionless runoff peak evaluated for the Bisagno – La Presa catchment. The maximum dimensionless runoff peak curve (bold line) is also reported together with the dimensionless hydrograph peaks (grey-filled stars) for the selected rainfall structure exponents ($n = 0.55, 0.62, 0.71$) and durations ($d_* = 0.5, 1.0, 1.5,$ and $2.0$).**

**Ref.#2 Comment C5:**

*Application. I do not understand the scope of applying the method to real events. In this case, if I want estimate the Qpeak, supposing to have a calibrated model I should use the rainfall time history, estimate the initial soil moisture and run the model to estimate Qpeak. The analysis done seems to me unuseful (but maybe because it is not clear the scope), what is the reason to build constant hyetograph for different durations picking the magnitude from a real event? You are building un-real rainfall events (and so un real catchment response...) when you already have the truth (..or a truth estimation).*

*If I well understand, in figure 10 a sort of DDF built with hr=80 mm (derived by the events, and which I suppose has a certain return period T\*) and n=0.39 (from Mediterranean statistical analysis) is compared with rainfall depth obtained by various n derived for each single event. But what does it mean this comparison? May be exist various H(T) > hr (for increasing T) that give same rainfall depths for the different durations but with n=0.39. Maybe the information is only that for those events, for some durations > basin response time (tr) the rainfall depth has a T larger than hr. I suggest: a) on one side better explaining the reasons and motivations of the presented experiment. b) On the other side I would like to see a sort of "project" experiment. So suppose to have the need to estimate the Q for a certain T, considering other methodologies (example Chicago hyetograph ? events with rainfall peaks at the end of hyetograph? Other?), and make a comparison. I think authors should start for the same data (DDF? Rainfall annual maxima on different duration?...) and compare the proposed method with other ones. Moreover I would introduce the effects of Soil Moisture*

Answer C5

Firstly, the authors would like to state clearly the scope of the catchment application section: the application should support the reader in the understanding of the proposed dimensionless approach. The catchment application is aimed to point out the dimensionless procedure implication and to provide some numerical examples of rainfall structure and their effects on the hydrograph peak (Figs. 10, 11 and 12 address graphically such task). As already mentioned, the general aim of the research is not to provide a hydrologic model to suitably estimate the hydrograph peak at given return periods or to verify the peak associated with an observed rainfall event. In the catchment application, the authors consider, as an example, three different rainfall events characterized by the same value of the maximum rainfall depth occurred at the reference time of the catchment ($h_r$= 80mm; $t_r$= 0.85 h) thus aiming to provide an example of three different rainfall structure according to the proposed approach. These three rainfall structures (i.e. *n* equal to 0.55, 0.62 and 0.71) represent only three of the possible outcomes in the sample space of the rainfall structures. Indeed, in Fig. 12 (see Fig. 12rev and 12new), the grey-filled stars are the dimensionless hydrograph peak resulting from input hyetograph characterized by the sampled *n* structure exponent values for the four selected dimensionless durations in the range [0.5, 2] where the structure exponent is assumed valid. In light of the previous consideration, the catchment application cannot be considered as a 'classical' verification of the proposed approach with experimental or numerical data.

It has to be noticed that a maximum rainfall depth at a given duration occurring in a specific catchment is characterized by a defined return period complying with the local DDF curves, however through the dimensionless procedure, the site-specific characteristics (such as the morphologic and climatic characteristics of the catchment) are no more relevant being included within the parameters of the dimensionless procedure (i.e. $h_r(T_r)$ and $t_r$) thus allowing to figure out the implication on the hydrograph peak irrespective of the absolute value of the rainfall depth (i.e. the corresponding return period). Even considering different initial soil moisture conditions (see Fig. 12new), the main findings illustrated in the contour plot of the dimensionless hydrograph peak are similar: the maximum

hydrograph peak tends to increase with increasing the rainfall structure exponent and the dimensionless rainfall duration while a saddle point is observed in the neighbourhood of $d_*$ and $n$ values equal to 1 and 0.3, respectively.

Finally, as documented in the literature, the classical iso-frequency assumption between the design rainfall event and the corresponding hydrograph peak is not strictly respected due to several factors including the influence of the initial moisture conditions on the resulting excess rainfall, the partial contributing area etc. (see e.g. Sivapalan et al., 1990), however, once again the author state that the determination of the flood frequency curve is out of the scope of the present research study.

The authors have revised the Introduction section (page 2, lines 16-18) and the Results and Discussion section (page 7, lines 14-15) in order to state clearly the scope of the catchment application section. The reviewed version of the text in the Introduction section is reported below and put in inverted commas.

"Finally a specific catchment application is discussed in order to point out the dimensionless procedure implications and to provide some numerical examples of the rainfall structures with respect to observed rainfall events; furthermore their effects on the hydrograph peak are examined."

The reviewed version of the text in the Results and Discussion section is reported below and put in inverted commas.

"Finally the dimensionless procedure is referred to a small Mediterranean catchment. In the catchment application the dimensionless procedure is fully specified as from the evaluation of the rainfall structures associated with three observed rainfall events as far as the determination of the reference peak flow and consequently of the dimensionless hydrograph peaks for the three observed rainfall structures."

Similarly the authors have revised the Section 3.3 "Catchment application" in order to better illustrate the catchment application purposes.

[revised manuscript text omitted]